# Orchestrating NK and T cells via tri-specific nano-antibodies for synergistic antitumor immunity

Qian-Ni Ye[1,2], Long Zhu[1], Jie Liang[1], Dong-Kun Zhao[1], Tai-Yu Tian[1], Ya-Nan Fan[1,3], Si-Yi Ye[1], Hua Liu[1], Xiao-Yi Huang[1], Zhi-Ting Cao[4], Song Shen ●[1,2] ✉ & Jun Wang ●[1,2,3] ✉

The functions of natural killer (NK) and T cells in innate and adaptive immunity, as well as their functions in tumor eradication, are complementary and intertwined. Here we show that utilization of multi-specific antibodies or nano-antibodies capable of simultaneously targeting both NK and T cells could be a valuable approach in cancer immunotherapy. Here, we introduce a tri-specific Nano-Antibody (Tri-NAb), generated by immobilizing three types of monoclonal antibodies (mAbs), using an optimized albumin/polyester composite nanoparticle conjugated with anti-Fc antibody. This Tri-NAb, targeting PDL1, 4-1BB, and NKG2A (or TIGIT) simultaneously, effectively binds to NK and CD8+ T cells, triggering their activation and proliferation, while facilitating their interaction with tumor cells, thereby inducing efficient tumor killing. Importantly, the antitumor efficacy of Tri-NAb is validated in multiple models, including patient-derived tumor organoids and humanized mice, highlighting the translational potential of NK and T cell co-targeting.

Natural killer (NK) and T5cells are essential for the innate and adaptive immune systems and play complementary and orchestrated roles in tumor recognition and eradication[1-4]. The unification of NK- and T-cell-based immunotherapies offers unprecedented opportunities[5,6]. Recent studies suggest that targeting checkpoint molecules, such as TIGIT (T-cell immunoreceptors with immunoglobulin and ITIM domains) and NKG2A (natural killer Group 2 member A), which are coexpressed on NK and T cells, can elicit concurrent innate and adaptive immunity[7-10]. However, monotherapy with the corresponding monoclonal antibodies (mAbs) has not achieved the desired outcomes in clinical trials due to the complexity of tumor pathogenesis, highlighting the need to combine mAbs with other checkpoint inhibitors[11,12]. Bi-/tri-specific antibodies (Bi-/Tri-Abs), which can target multiple antigenic epitopes, are promising options for combination therapies[13-15]. Bi-Abs have significantly broadened therapeutic options for treating tumors, with twelve already approved as of July 2023. Tri-Abs represent a further

development in this area, with their three distinct specificities that facilitate the simultaneous engagement of different immunomodulatory molecules with enhanced flexibility[14,16-18]. We propose that through careful selection, Tri-Abs capable of targeting NK and T cells could achieve favorable clinical outcomes; however, no such Tri-Abs have been reported to date.

The inadvertent generation of byproducts poses significant challenges to the manufacturing process of Tri-Abs, resulting in time-consuming procedures, high costs, and low yields[19,20]. Moreover, their application is impeded by postproduction issues such as degradation, aggregation, and fragmentation[21,22]. The immobilization of multiple mAbs on the surface of nanocarriers is a promising alternative for constructing multi-specific nano-antibodies (NAbs). Bi-/tri-specific NAbs with remarkable antitumor effects have been reported[23-26]; however, their complex manufacturing process and potential for impaired antibody function impede their clinical implementation[27,28].

[1]School of Biomedical Sciences and Engineering, South China University of Technology, Guangzhou International Campus, Guangzhou, P. R. China. [2]National Engineering Research Center for Tissue Restoration and Reconstruction, South China University of Technology, Guangzhou, P. R. China. [3]Guangdong Provincial Key Laboratory of Biomedical Engineering, South China University of Technology, Guangzhou, P. R. China. [4]School of Biopharmacy, China Pharmaceutical University, Nanjing, P. R. China. ✉e-mail: shensong@scut.edu.cn; mcjwang@scut.edu.cn

We previously engineered a mAbs nano-anchor (mAbs-NA) by conjugating anti-IgG (Fc) antibodies (αFc) to aminated polystyrene nanoparticles to address these challenges, which could immobilize mAbs and serve as a platform for multispecific NAbs[29,30]. Nonetheless, the poor biocompatibility of polystyrene cores reduces their efficiency and impairs their clinical use. In this study, we generate an albumin/polyester composite nanoparticle (APCN) to replace the polystyrene core of mAbs-NA, addressing biosafety concerns. Furthermore, we immobilize three types of mAbs targeting PDL1 (a well-known checkpoint protein and tumor-associated antigen)[31], NKG2A (an inhibitory member of the NKG2 family expressed in activated NK and T cells)[32], and 4-1BB (a costimulatory glycoprotein receptor expressed in activated NK and T cells)[33] on APCN-based mAbs-NAs (APCN@NAs) and develop a tri-specific Nano-Antibody (Tri-NAb). The Tri-NAb has remarkable antitumor efficacy in vitro and in vivo by orchestrating NK and T cells. Notably, the engineered humanized PDL1/4-1BB/TIGIT Tri-NAb shows encouraging therapeutic effects on patient-derived tumor organoids and humanized nice, highlighting the translational potential of synergistic NK and T cell (Fig. 1).

## Results

### Construction, optimization, and characterization of APCNs

The mAbs-NA represent an exceptional platform for the development of multispecific NAbs[29]; our initial focus was to create a nanoparticle with good biosafety to replace the polystyrene core prior to constructing the Tri-NAb targeting NK and T cells. Serum albumin (SA) is widely employed in drug delivery because of its excellent biocompatibility, prolonged circulatory half-life, and abundant modifiable groups, such as amino and sulfhydryl groups. Notably, SA possesses hydrophobic pockets with a strong affinity for binding long-chain fatty acids (LCFAs) and drugs[34–36]. Albumin-bound paclitaxel (Abraxane®, nab-Paclitaxel), a well-known nanomedicine, is produced by binding paclitaxel to hydrophobic pockets via high-pressure homogenization. Inspired by this approach, we hypothesized that hydrophobic LCFAs or polyesters could form stable nanoparticles by binding to albumin, creating a distinct alternative to Abraxane®, which tends to disintegrate upon injection[37]. Thus, the abundant amino groups on albumin can be conveniently harnessed for coupling an αFc to construct next-generation mAbs-NAs. We formulated stearic acid (a type of LCFA) into nanoparticles (LCFA-NPs) with bovine serum albumin (BSA) using a single ultrasonic emulsification and solvent evaporation method to verify this hypothesis. The highly variable size and polydispersity index (PDI), as determined by dynamic light scattering (DLS), indicated the unsatisfactory stability of the LCFA-NPs in water and PBS. (Supplementary Fig. 1b). Scanning electron microscopy (SEM) and transmission electron microscopy (TEM) images revealed a lack of uniformity and an irregular structure of the LCFA-NPs (Supplementary Figs. 1c, d), which failed to improve even after replacing stearic acid with palmitic acid. Consequently, we explored the use of biodegradable hydrophobic polyesters as an alternative solution. Molecular dynamics simulations demonstrated that poly(L-lactide) (PLLA), a representative polyester, was intricately wrapped and

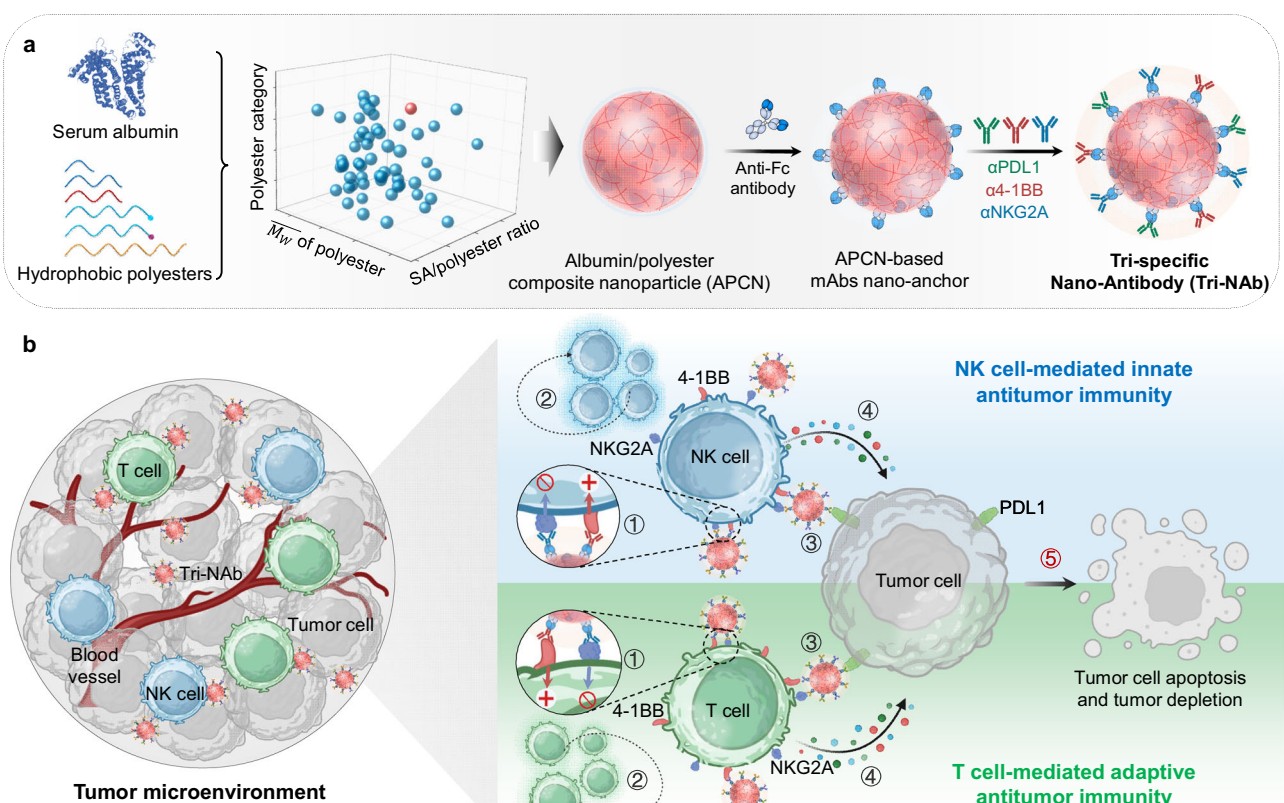

**Fig. 1 | Schematic diagram illustrating the construction of the Tri-NAb and its proposed mechanism of action to potentiate both innate and adaptive antitumor immune responses. a** Initially, a biocompatible albumin/polyester composite nanoparticle (APCN) with the optimal formulation was meticulously engineered through multivariate screening. Subsequently, the tri-specific Nano-Antibody (Tri-NAb) was acquired by immobilizing αPDL1/α4-1BB/αNKG2A onto APCN preassembled with αFc (APCN@NA). **b** Following administration, the Tri-NAb accumulated at the tumor site, effectively triggering the activation (①) and proliferation (②) of natural killer (NK) and CD8+ T cells while augmenting their interactions with tumor cells (③), thereby stimulating the release of cytotoxic granules (④). This orchestrated interplay culminated in the induction of efficient tumor cell apoptosis (⑤), harnessing synergistic effects of innate and adaptive immunity mediated by NK and CD8+ T cells. Figure 1b was created with BioRender.com under a CC-BY-NC-ND license.

partially embedded in human serum albumin (HSA) to form a complex (Fig. 2a and Supplementary Fig. 2a). Notably, the positively charged regions of HSA were predominantly exposed on the surface of the complex, facilitating further modifications (Supplementary Fig. 2b). The calculations indicated a theoretical binding free energy between PLLA and HSA of −1766.291 kJ/mol, indicating their stable combination through multiple interactions (Fig. 2b). The primary contributors to this stability included negative energy terms, such as electrostatic energy ($\Delta$Eelec: −225.117 ± 28.470 kJ/mol), van der Waals energy ($\Delta$EvdW: −3257.829 ± 55.833 kJ/mol), and nonpolar solvation energy ($\Delta$Gnp, solv: −332.512 ± 6.276 kJ/mol), while positive electrostatic solvation energy ($\Delta$Gelec, solv: 2049.167 ± 81.321 kJ/mol) had a minimal impact (Fig. 2b).

As a method to optimize the efficiency of albumin loading, a library of albumin/polyester nanoparticles was constructed to screen for the optimal formulation by varying the type of polyester, weight-average molecular weight ($\overline{M_w}$), and weight ratio of polyester to SA (Fig. 2c, Supplementary Figs. 3 and 4). The SA loading efficiency of selected NPs with a size range of 100−150 nm and a PDI of 0.01−0.02 was measured using high-performance liquid chromatography (HPLC) (Figs. 2d, e). The top six NPs with high BSA-loading efficiencies were further prepared and quantified using HSA instead of BSA, confirming a similarly high HSA-loading efficiency (Supplementary Fig. 5). TEM analysis revealed that the NPs prepared using Formula 4 exhibited the most uniform and compact spherical structure, whereas a minimal free HSA content was detected after purification by high-speed centrifugation, as determined using HPLC analysis (Fig. 2f and Supplementary Fig. 6). Therefore, a formulation composed of PLLA with an ester end group (PLLA-COOR, R = -$(CH_2)_{11}CH_3$, $\overline{M_w}$ = 137 K) and HSA with a weight ratio of 1/10 was chosen as the optimal APCN for further investigation. The fluorescence spectrum revealed significant attenuation of the intrinsic fluorescence signal from HSA upon the addition of PLLA, suggesting that the hydrophobic environment surrounding the chromophore molecules may have been altered by PLLA (Supplementary Fig. 7). Stochastic optical reconstruction microscopy (STORM) images revealed the colocalization of PLLA and HSA within APCNs after labeling with rhodamine B (PLLA) and iFluor™ 488 (HSA) (Fig. 2g). TEM imaging confirmed the presence of APCNs with a size of approximately 100 nm, while the presence of HSA was further validated by elemental mapping of C, N, O, and S (Fig. 2h). The size and zeta potential of APCNs in a buffer with a pH of 2−13 fluctuated, which was also attributed to the presence of the loaded HSA (Supplementary Fig. 8). Encouragingly, even after an incubation in water, PBS, or FBS for six days, as observed using SEM analysis, the APCNs exhibited minimal variations in size, indicating excellent stability under in vitro conditions (Supplementary Fig. 9).

## Construction and characterization of the Tri-NAb

The abundant amino groups on APCN, with a density of approximately 4.0/nm², were demonstrated to be capable of coupling $\alpha$Fc and constructing the mAb-NA (Fig. 2i and Supplementary Fig. 10). The enzyme-linked immunosorbent assay (ELISA) results showed that over 70% of the $\alpha$Fc was effectively immobilized onto APCN when the weight ratio of APCN to $\alpha$Fc was equal to or exceeded 12:1 (Supplementary Fig. 11). After the conjugation of $\alpha$Fc, the average hydrodynamic diameter of APCN increased by approximately 20 nm (Fig. 2j), while the zeta potential shifted from −18.1 to −34.6 mV. SEM images revealed a transformation from a smooth spherical surface to a rough texture, indicating the successful immobilization of $\alpha$Fc on APCN (Figs. 2j, k). After eight days of incubation in water or PBS, minimal changes in size were observed for APCN@NA, demonstrating its remarkable stability in vitro (Fig. 2l and Supplementary Fig. 12). Preliminary immune electron microscopy imaging revealed that gentle mixing enabled the immobilization of an anti-PDL1 antibody ($\alpha$PDL1, depicted as black dots) on the surface of APCN@NA (Supplementary Fig. 13).

The expression of 4-1BB and NKG2A on activated NK and CD8⁺ T cells has been reported[5] and was also confirmed in our experiments (Supplementary Fig. 14); however, the combined immunological effect remains unclear. In this study, we devised a Tri-NAb by simultaneously immobilizing $\alpha$PDL1, anti-4-1BB ($\alpha$4-1BB), and anti-NKG2A ($\alpha$NKG2A) mAbs on APCN@NA. Tri-NAbs can recognize PDL1-expressing tumor cells and synergistically activate NK and CD8⁺ T cells for effective tumor eradication. The dissociation constants (Kd values) of APCN@NA for rat IgG2a and IgG2b were 0.70 ± 0.13 and 0.41 ± 0.32 nM, respectively, according to microscale thermophoresis (MST). This result indicates that APCN@NA exhibited strong affinity for the three selected antibodies ($\alpha$PDL1: IgG2a; $\alpha$4-1BB: IgG2a; and $\alpha$NKG2A: IgG2b) (Figs. 2m, n). The Tri-NAb was then obtained by gently mixing APCN@NA with these antibodies, resulting in a visually striking spherical structure observed in SEM images, with an average hydrodynamic diameter of 170 nm and zeta potential of −10.0 mV (Supplementary Fig. 15). By labeling the antibodies with different fluorescent dyes, the simultaneous detection of all three fluorescent molecules confirmed their adherence to Tri-NAbs at a final ratio of approximately 1:1:1, along with excellent stability in vitro (Figs. 2o, p, and Supplementary Fig. 16). Additionally, APCN@NA loaded with $\alpha$PDL1 and $\alpha$4-1BB (NP$_{\alpha PDL1+\alpha4-1BB}$), APCN@NA loaded with $\alpha$PDL1 and $\alpha$NKG2A (NP$_{\alpha PDL1+\alpha NKG2A}$), and APCN@NA loaded with $\alpha$4-1BB and $\alpha$NKG2A (NP$_{\alpha4-1BB+\alpha NKG2A}$) were prepared for subsequent comparisons using the same method described above.

## The Tri-NAb effectively activated both NK and CD8⁺ T cells in vitro

ELISA was performed to validate the ability of the therapeutic antibodies to bind to the corresponding antigens, and the results indicated that the Tri-NAb exhibited an affinity nearly indistinguishable from that of free $\alpha$PDL1, $\alpha$4-1BB, and $\alpha$NKG2A, indicating the applicability of this antibody anchoring approach (Fig. 3a). We initially investigated the ability of the Tri-NAb to bind to tumor, NK, and T cells to evaluate its functionality at the cellular level. Flow cytometry analyzes revealed that the fluorescein isothiocyanate (FITC)-labeled Tri-NAb effectively adhered not only to tumor cells with high PDL1 expression but also to NK and CD8⁺ T cells expressing both 4-1BB and NKG2A (Figs. 3b−d, Supplementary Figs. 17 and 14). After extracellular fluorescence quenching using 0.4% trypan blue, a faint fluorescence signal was detected, suggesting that the Tri-NAb was firmly bound to the cell surface rather than internalized within the cell (Figs. 3b−d). When NK/T cells were preincubated with excess $\alpha$4-1BB and $\alpha$NKG2A or when tumor cells were incubated with excess $\alpha$PDL1, Tri-NAb binding to these cells was significantly decreased, indicating a high likelihood of Tri-NAb binding through targeting with mAbs (Supplementary Fig. 18). Compared with free $\alpha$4-1BB, Tri-NAb remarkably induced the proliferation of NK and CD8⁺ T cells, which can be attributed to the presence of multiple $\alpha$4-1BB molecules on the surface of Tri-NAb and the multimerization of 4-1BB on NK and T cells treated with Tri-NAb[38,39] (Figs. 3e, f, and Supplementary Fig. 19). A coculture experiment was conducted to further explore the role of Tri-NAbs in the interaction between tumors and NK/CD8⁺ T cells (Fig. 3g). Confocal laser scanning microscopy (CLSM) images revealed that both the NP$_{\alpha PDL1+\alpha4-1BB}$- and Tri-NAb-treated groups exhibited enhanced interactions between CD8⁺ T (green) and MC38 tumor cells (red), while NP$_{\alpha PDL1+\alpha NKG2A}$ and Tri-NAb effectively promoted interactions between NK (blue) cells and tumor cells. However, these phenomena were scarcely observed in the control group, the free $\alpha$PDL1 & $\alpha$4-1BB & $\alpha$NKG2A (Tri-mAbs) group, or the NP$_{\alpha4-1BB+\alpha NKG2A}$ group lacking $\alpha$PDL1, which was presumably due to the decreased recognition and binding capacity toward tumor cells (Supplementary Fig. 20). Evidently, the Tri-NAb played dual roles, as both NP$_{\alpha PDL1+\alpha4-1BB}$ and NP$_{\alpha PDL1+\alpha NKG2A}$ not

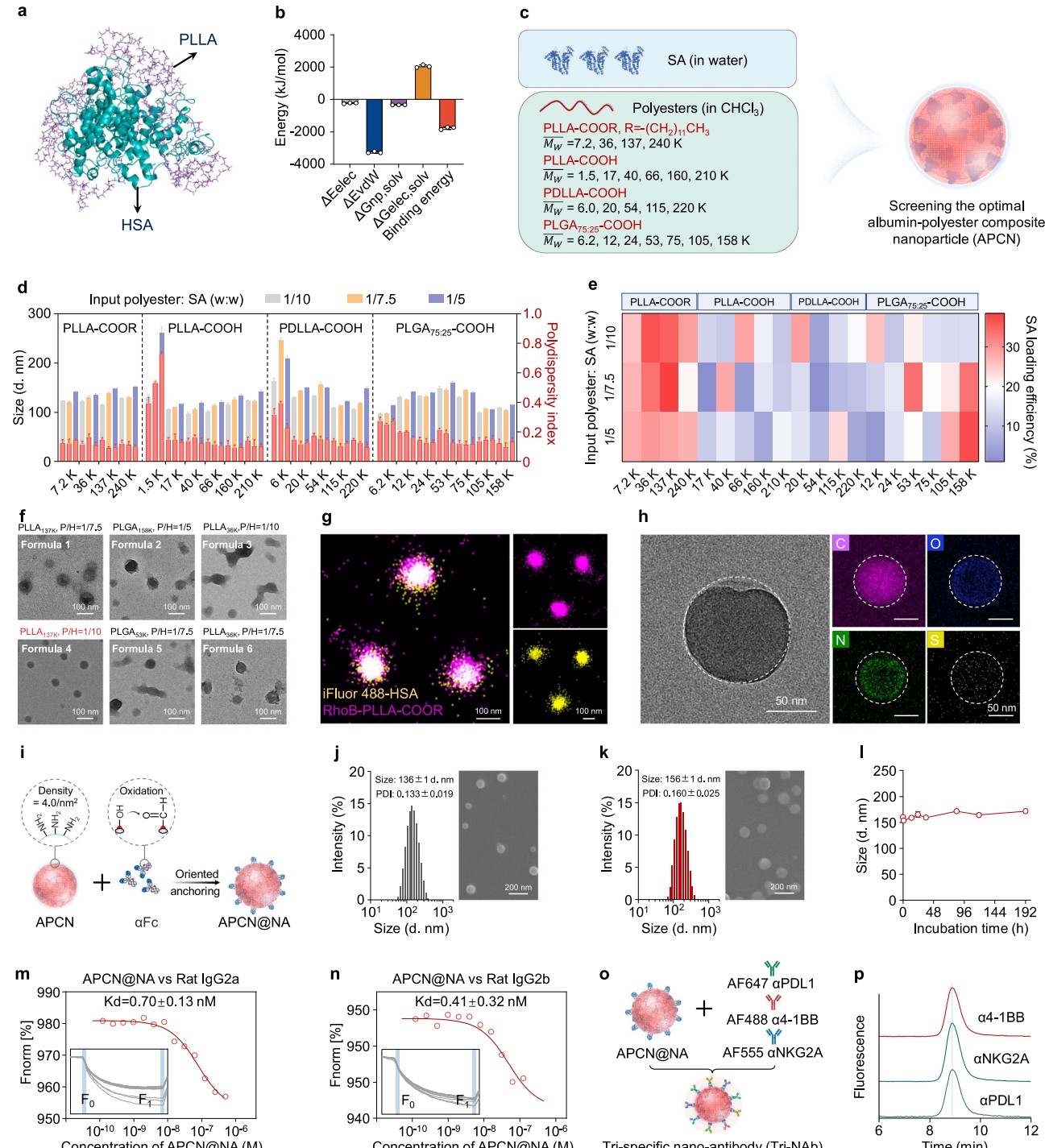

**Fig. 2 | Construction and characterization of APCN and Tri-NAb. a** Molecular dynamics simulation pattern. **b** Binding free energy and individual energy terms. Data are presented as means ± s.d. ($n = 3$ independent calculation). **c** Scheme of the category and molecular weight ($\overline{M_W}$) of polyesters for screening the optimal formula. **d** Average hydrodynamic size and PDI of NPs prepared using different formulas as determined by DLS. Data are presented as means ± s.d. ($n = 5$ biologically independent samples). **e** HPLC analysis of the SA-loading efficiency of NPs prepared using different formulations. **f** Representative TEM images of NPs prepared with the top six formulations with the highest SA loading efficiency. P/H is the weight ratio of polyester to HSA. **g** STORM images of APCN. HSA and PLLA were labeled with iFluor 488 and RhoB, respectively. **h** Representative TEM image and EDS elemental map (C, N, O, and S) of APCN. **i** Scheme of the construction of APCN@NA. αFc was oxidized and immobilized onto HSA contained in APCN through an aldehyde−amine reaction. Size distribution and representative SEM images of APCN (**j**) and APCN@NA (**k**). Data are presented as means ± s.d. ($n = 6$ biologically independent samples). **l** Size variation of APCN@NA in PBS during incubation for eight days. Data are presented as means ± s.d. ($n = 5$ biologically independent samples). **m**, **n** MST spectra indicating that APCN@NA has a similar affinity for rat IgG2a and IgG2b, namely, the three selected antibodies (αPDL1, α4-1BB, and αNKG2A). **o** Scheme of the construction of the Tri-NAb through gentle mixing of APCN@NA and the three antibodies (αPDL1, α4-1BB, and αNKG2A). **p** HPLC-FLR confirmed that the three antibodies were immobilized on the Tri-NAb. Source data are provided as a Source Data file.

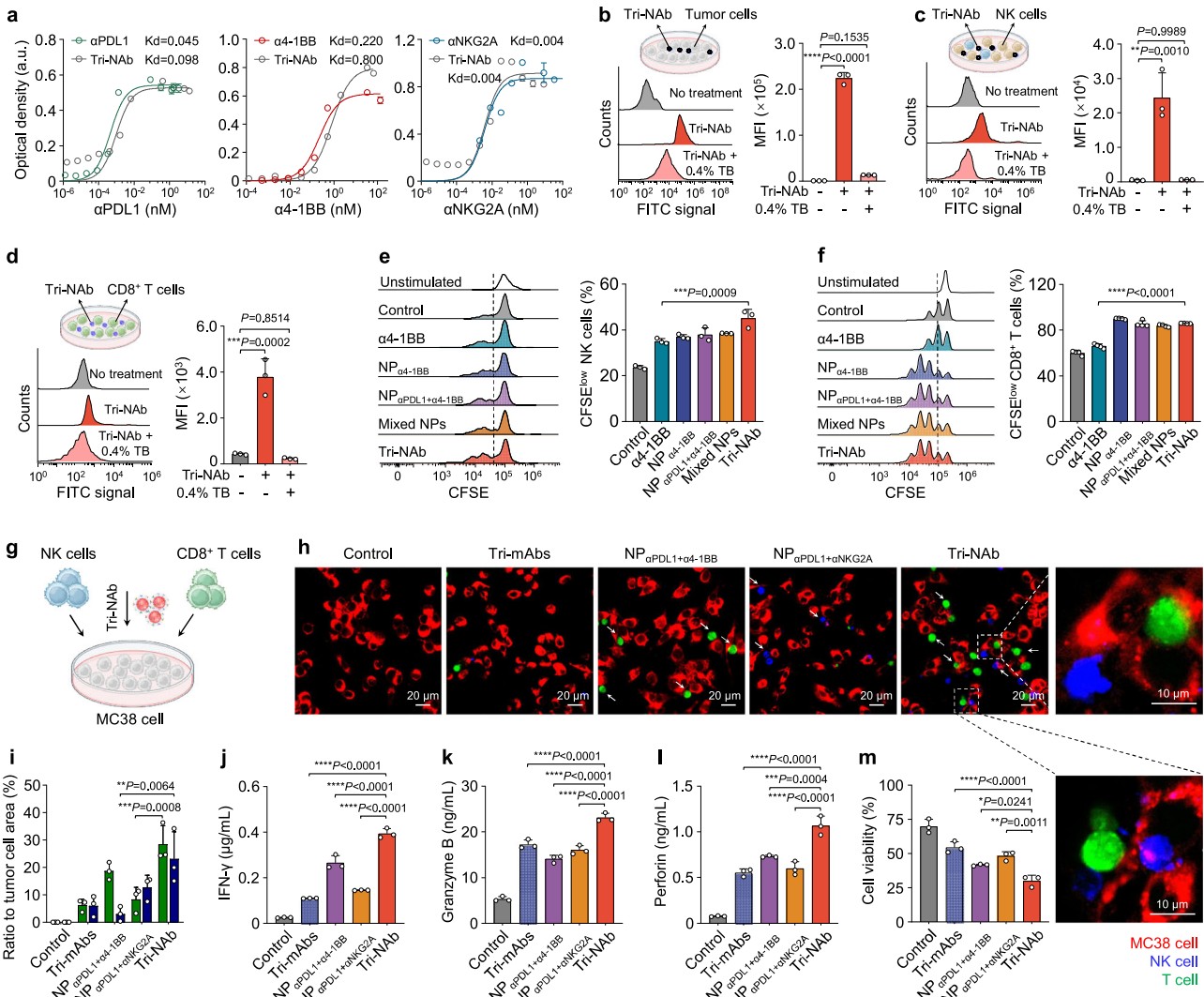

**Fig. 3 | Tri-NAb effectively activated both NK and CD8⁺ T cells in vitro. a** Capacity of Tri-NAb to bind recombinant mouse PDL1, 4-1BB, and NKG2A, as determined using ELISAs. Data are presented as means ± s.d. ($n = 3$ biologically independent samples). Representative FACS histograms of MC38 murine colon cancer cells (**b**) NK cells (**c**) and CD8⁺ T cells (**d**) after an incubation with PBS, FITC-labeled Tri-NAb, or FITC-labeled Tri-NAb with 0.4% trypan blue. Data are presented as means ± s.d. ($n = 3$ biologically independent samples). MFI, median fluorescence intensity. n.s., not significant. Representative FACS histograms of CFSE-labeled NK cells (**e**) and CFSE-labeled CD8⁺ T cells (**f**) after an incubation with 1) unstimulated, 2) control, 3) α4-1BB, 4) NP$_{α4-1BB}$, 5) NP$_{αPDL1+α4-1BB}$, 6) NP$_{αPDL1+α4-1BB}$ & NP$_{αPDL1+αNKG2A}$ (Mix-NPs), or 7) Tri-NAb. Data are presented as means ± s.d. (NK cells: $n = 3$ biologically independent samples, CD8⁺ T cells: $n = 4$ biologically independent samples). **g** Scheme of the interaction and cytotoxicity of NK and CD8⁺ T cells toward MC38 tumor cells after an incubation with 1) control, 2) αPDL1 & α4-1BB & αNKG2A (Tri-mAbs), 3) NP$_{αPDL1+α4-1BB}$, 4) NP$_{αPDL1+αNKG2A}$, or 5) Tri-NAb. **h** Representative confocal images

of NK and CD8⁺ T cells interacting with MC38 tumor cells after treatment with the different formulations for 6 h. NK cells were labeled with DiD dye (blue); CD8⁺ T cells were labeled with CFSE (green); and MC38 cells were labeled with PKH26 dye (red). **i** The proportions of NK or CD8⁺ T cells in the visual field were evaluated by computing the green or blue signal ratio in the merged image. Data are presented as means ± s.d. ($n = 3$ nonoverlapping images). The calculation was normalized to the red signal (tumor cell). The levels of IFN-γ (**j**), granzyme B (**k**), and perforin (**l**) in the supernatant of the coincubation system were examined using ELISAs. **m** Viability of MC38-luc cells in the coincubation system. MC38-luc, MC38 cells expressing the luciferase gene. Data are presented as means ± s.d. ($n = 3$ biologically independent samples). Statistical significance was determined using one-way ANOVA with Tukey's post hoc test. *$P < 0.05$; **$P < 0.01$; ***$P < 0.001$; and ****$P < 0.0001$. Figure 3b–d and g were created with BioRender.com under a CC-BY-NC-ND license. Source data are provided as a Source Data file.

only effectively promoted bridging between NK cells and tumor cells but also facilitated bridging and interactions between CD8⁺ T cells and tumor cells (Fig. 3h). These interactions were further confirmed by calculating the bridging rate and intercellular distance, as well as the formation of immune synapses, as shown in Fig. 3i and Supplementary Figs. 21, 22. Moreover, the Tri-NAb-treated group exhibited significant increases in the levels of IFN-γ, perforin, and granzyme B released by NK and CD8⁺ T cells in the coculture system (Figs. 3j–l), resulting in an intensified killing effect on MC38-luc cells (Fig. 3m). However, the levels of these cytokines released by the NP$_{α4-1BB+αNKG2A}$-treated group and their

tumor-killing activity were not as robust as those of the Tri-NAb-treated group, highlighting the importance of αPDL1-mediated tumor cell recognition and bridging for Tri-NAb to exert excellent antitumor efficacy (Supplementary Fig. 23). Except for that group, the viability of tumor cells within 24 h was monitored using a high-content analysis (HCA) platform (Supplementary Fig. 24). Additionally, the Tri-NAb augmented the activation of downstream signaling pathways of NK/T cells, resulting in cytokine release and tumor cell killing when NK or CD8⁺ T were cultured alone with tumor cells, highlighting the crucial importance of both NK and CD8⁺ T cells (Supplementary Figs. 25–27). Although NK/T cells also

express PDL1 and can associate with the Tri-NAb, we did not observe any differences in fratricidal behavior between NK and CD8[+] cells when they were cultured with Tri-NAb (Supplementary Video 1–3). This result may be attributed to the relatively weak cellular interaction of NK/T cells facilitated by Tri-NAb, as well as their inherent self-protective mechanism. The aforementioned findings confirmed that Tri-NAb facilitated the substantial activation and proliferation of NK and CD8[+] T cells, augmented the interaction between NK/CD8[+] T cells and tumor cells, and strengthened innate and adaptive antitumor immune responses.

## Tri-NAb effectively eradicated colon cancer in vivo and enhanced both innate and adaptive immune responses

Encouraged by the enhanced activation of NK and CD8[+] T cells mediated by Tri-NAb in vitro, we explored the antitumor activity of Tri-NAb in vivo. First, we evaluated the pharmacokinetic properties of plasma. The detection of the fluorescence intensity of $\alpha$PDL1 revealed a significant increase in the circulation half-life of the mAbs (T1/2, 12.27 h) when conjugated to APCN@NA (NP$_{\alpha PDL1}$, T1/2, 49.50 h), indicating a remarkable extension of the residence time of the mAbs (Fig. 4a). Subsequently, we examined the in vivo biodistribution of Tri-NAb. Ex vivo fluorescence imaging revealed intense fluorescence in the tumor tissue 36 h after the administration of Cy5-labeled Tri-NAb, suggesting greater accumulation of these antibodies than of Cy5-labeled Tri-mAbs (Figs. 4b, c). Immunofluorescence staining and imaging of tumor tissue sections revealed that the penetration of Tri-NAb, which is equipped with therapeutic antibodies, into tumor tissues was significantly increased compared to NP$_{IgG}$, while Tri-NAb effectively mediated the interaction between NK/T cells and tumor cells in vivo, all of which were conducive to the better antitumor efficacy of Tri-NAb (Supplementary Figs. 28 and 29). We used an MC38 murine colon cancer model to assess the therapeutic potential of Tri-NAbs in vivo (Fig. 4d). Compared to the untreated group, Tri-mAbs demonstrated moderate antitumor activity, resulting in a 4-day extension in the median survival of mice compared to the untreated group (46 vs. 50 days). NP$_{\alpha PDL1+\alpha4-1BB}$ induced favorable tumor suppression and significantly prolonged the median tumor survival to 66 days compared with 52 days in the NP$_{\alpha PDL1+\alpha NKG2A}$ treatment group. Remarkably, treatment with Tri-NAb markedly suppressed tumor growth and prolonged survival to 72.5 days. Notably, three mice achieved complete tumor remission (Figs. 4e–g), and no significant weight variations were observed during treatment (Supplementary Fig. 30). Furthermore, the ability of Tri-NAb to inhibit initial tumor growth in subcutaneous MC38-luc murine colon cancer mice was evaluated using an in vivo imaging system (IVIS). The IgG control, Tri-mAbs, or Tri-NAb was intravenously administered when the tumor volume reached approximately 150 mm³. The gradual regression or even complete eradication of tumors following Tri-NAb treatment (2/5 of mice), as indicated by the visible reduction or absence of bioluminescence, contrasted with the continued growth after Tri-mAbs treatment (Figs. 4h–j). A similar phenomenon was observed in the tumor model with an initial volume of 300 mm³ (Figs. 4k–m), further highlighting the exceptional antitumor potential of Tri-NAb. Importantly, our injectable doses of APCN and Tri-NAb exhibited good safety profiles in healthy mice, as indicated by serum biochemical and histological analyses, compared with the group treated with free antibodies (Tri-mAbs), which showed some degree of hepatic or pulmonary injury (Supplementary Fig. 31). Moreover, no significant side effects were observed on the main organs of the tumor-free mice at 2 months after treatment (Fig. 4n).

Variations in the immune system were further investigated by conducting mRNA sequencing on the tumor tissues from treated mice.

Sequence reads were obtained using the DNBSEQ platform in paired-end mode (150 bp), checked and filtered using SOAPnuke software to obtain clean reads, which were subsequently aligned to the relevant mouse reference genome version. The analysis revealed 914 differentially expressed genes (DEGs) between the Tri-mAbs and Tri-NAb groups (Fig. 5a). Gene Ontology (GO) analysis of the biological process annotations of DEGs in the Tri-NAb group indicated that these genes were primarily associated with immune system processes, immune responses, and adaptive and innate immune response signaling pathways (Fig. 5b). Gene set enrichment analysis (GSEA) revealed a trend toward upregulation in DEGs enriched in innate and adaptive immune response pathways after treatment with Tri-NAb (Figs. 5c, d). Among the genes involved in NK cell-mediated cytotoxicity, T-cell activation, and T-cell proliferation signaling pathways, higher expression levels were observed in the Tri-NAb treatment group than in the other groups (Figs. 5e–g). Overall, these findings confirm that by coordinating innate and adaptive immunity involving NK and CD8[+] T cells, Tri-NAb effectively inhibits MC38 colon cancer growth at various stages.

Specific antibodies were selected to deplete the NK and CD8[+] T cell populations to further evaluate their involvement in Tri-NAb-mediated antitumor immune responses (Supplementary Fig. 32a). The depletion of CD8[+] T cells significantly impeded the remarkable antitumor efficacy of Tri-NAb, as evidenced by the tumor growth curves; similarly, the depletion of NK cells had a substantial impact on the antitumor effect of Tri-NAb, particularly on the complete remission rate (CR). As anticipated, simultaneous depletion of both CD8[+] T and NK cells mediated by anti-CD8 and anti-NK1.1 antibodies resulted in a complete loss of the antitumor potency exhibited by Tri-NAb (Supplementary Fig. 32b–d). These findings underscore the crucial roles played by both NK and CD8[+] T cells in mediating the anticancer response elicited by Tri-NAbs, with CD8[+] T cells being predominant. However, in T- and B-cell-null Rag1-deficient (Rag1$^{-/-}$) mice, colon cancer growth was mildly inhibited following Tri-NAb treatment (Supplementary Fig. 33), but this effect was not as pronounced as that observed in immunocompetent C57BL/6 mice (Fig. 4f). These findings likewise suggest that the antitumor response elicited by Tri-NAb is moderately reliant on NK cells, with T cells playing a more crucial role.

## Tri-NAb effectively inhibited the progression of melanoma in vivo by simultaneously promoting the infiltration of NK and CD8+ T cells

We evaluated the therapeutic efficacy of the Tri-NAb in a murine melanoma model to investigate its applicability as a viable therapeutic modality (Fig. 6a). Similarly, compared with the IgG control and Tri-mAbs, treatment with NP$_{\alpha PDL1+\alpha4-1BB}$, NP$_{\alpha PDL1+\alpha NKG2A}$, or NP$_{\alpha4-1BB+\alpha NKG2A}$ exhibited certain advantages in tumor control, while Tri-NAb demonstrated superior inhibition of tumor progression (Figs. 6b, c and Supplementary Fig. 34). After the administration of three doses, we analyzed immune cell infiltration in the tumor tissue using flow cytometry (Supplementary Fig. 35). The percentages and numbers of both NK and CD8[+] T cells were markedly greater in the Tri-NAb-treated group than in the other treatment groups (Figs. 6d–i and Supplementary Fig. 34), indicating that Tri-NAb effectively activated NK and CD8[+] T cells, leading to enhanced antitumor activity. Moreover, the survival time of the mice was significantly prolonged, with 2 of 9 mice achieving complete tumor regression (Fig. 6j), while no significant changes in mouse weight or inflammatory cytokine levels in the peripheral blood were observed during treatment (Supplementary Figs. 36 and 37). We halved the dose of $\alpha$4-1BB to 6.25 mg/kg to treat mice with subcutaneous melanoma, as described above, to further minimize drug toxicity (Fig. 6a). Melanoma cell growth was effectively inhibited with an efficacy comparable to that of Tri-NAb treatment alone. Notably, even at

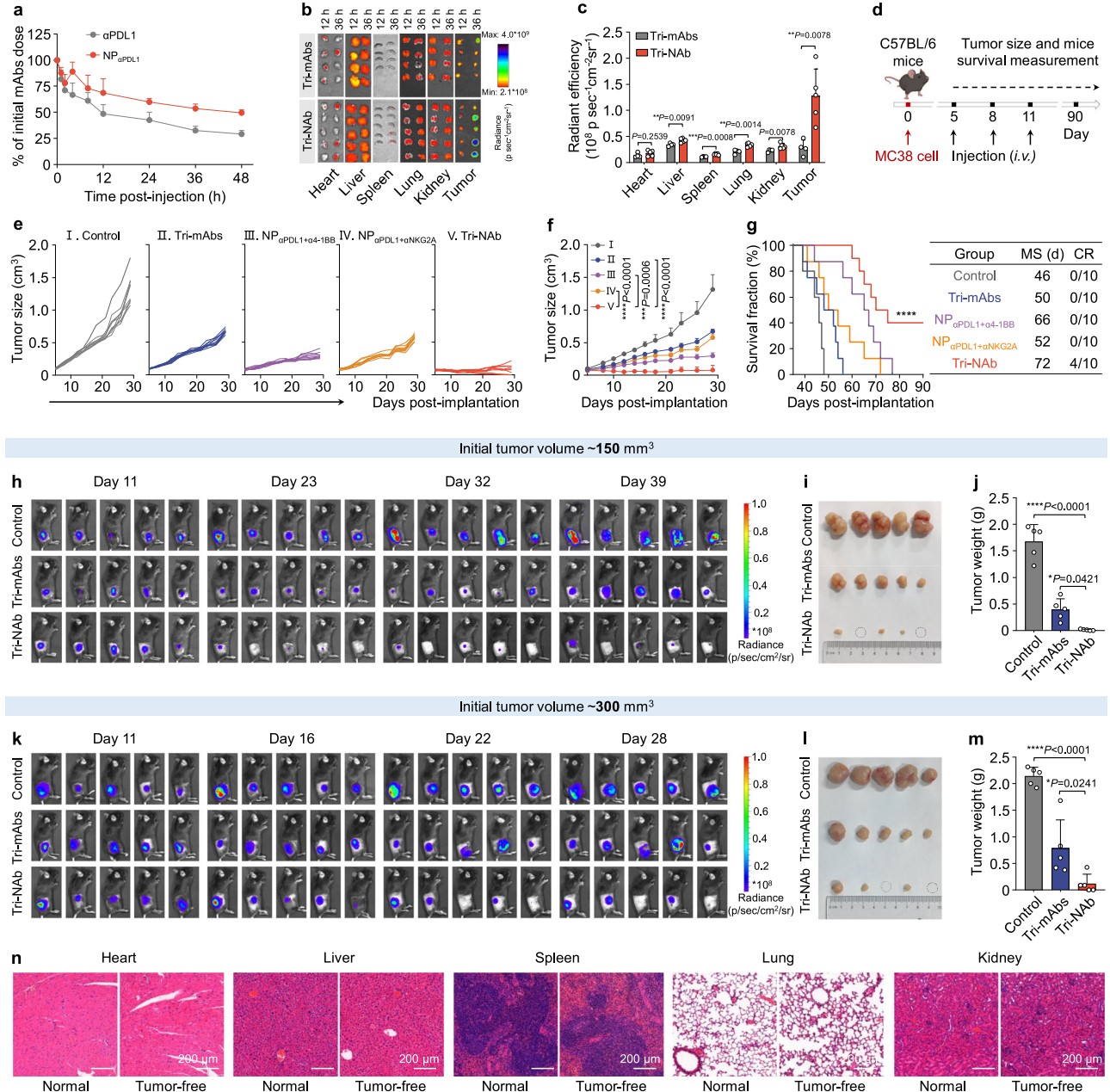

**Fig. 4 | Tri-NAb effectively eradicated murine colon cancer in vivo. a** Blood circulation half-life of αPDL1 and NP$_{αPDL1}$ evaluated in female C57BL/6 mice using Cy5-labeled αPDL1. **b** Ex vivo fluorescence imaging. Major organs and tumor tissues of female C57BL/6 mice were collected for ex vivo imaging at 12 and 36 h after the i.v. injection of Cy5-labeled Tri-mAbs or Tri-NAb. **c** Fluorescence quantification of Cy5-labeled Tri-mAbs (represented by gray bars) or Tri-NAb (represented by orange bars) in organs and tumor tissues. The Tri-mAbs data are presented as means ± s.d. ($n = 4$ biologically independent mice). The Tri-NAb data are presented as means ± s.d. ($n = 5$ biologically independent mice). Statistical significance was calculated using two-way ANOVA followed by the Bonferroni test. *$P < 0.05$; **$P < 0.01$. **d** Experimental scheme of the subcutaneous MC38 murine colon cancer model in female C57BL/6 mice. Different formulations with equivalent doses of αPDL1, α4-1BB, and αNKG2A (2.5 mg/kg each) were i.v. administered via the tail vein on Days 5, 8, and 11 after MC38 tumor cell inoculation. Individual (**e**) and average (**f**) tumor growth curves of MC38 tumors in animals treated with different formulations. Data are presented as means ± s.d. ($n = 10$ mice per group). **g** Survival curves for each group of mice. MS, median survival time. CR, complete response. Data are presented as means ± s.d. ($n = 10$ mice per group). In vivo bioluminescence imaging showed the therapeutic effect of Tri-mAbs or Tri-NAb on the MC38-luc tumor model in female C57BL/6 mice with an initial volume of 150 (**h**) or 300 mm³ (**k**). The equivalent injection doses of αPDL1, α4-1BB, and αNKG2A were 2.5 mg/kg. **i–l** Tumor images collected from treatment endpoints. The solid black circles indicate tumor-free tissues after treatment. **j, m** Tumor weights after different treatments. Data are presented as means ± s.d. ($n = 5$ mice per group). **n** Representative images of H&E staining of major organs in normal and tumor-free mice 2 months after the end of (**d**) treatment. For **g** statistical significance was calculated using the log-rank (Mantel-Cox) test. For **c, j,** and **m**, statistical significance was calculated using one-way ANOVA with Tukey's post hoc test. *$P < 0.05$; **$P < 0.01$; ***$P < 0.001$; and ****$P < 0.0001$. Figure 4d was created with BioRender.com under a CC-BY-NC-ND license. Source data are provided as a Source Data file.

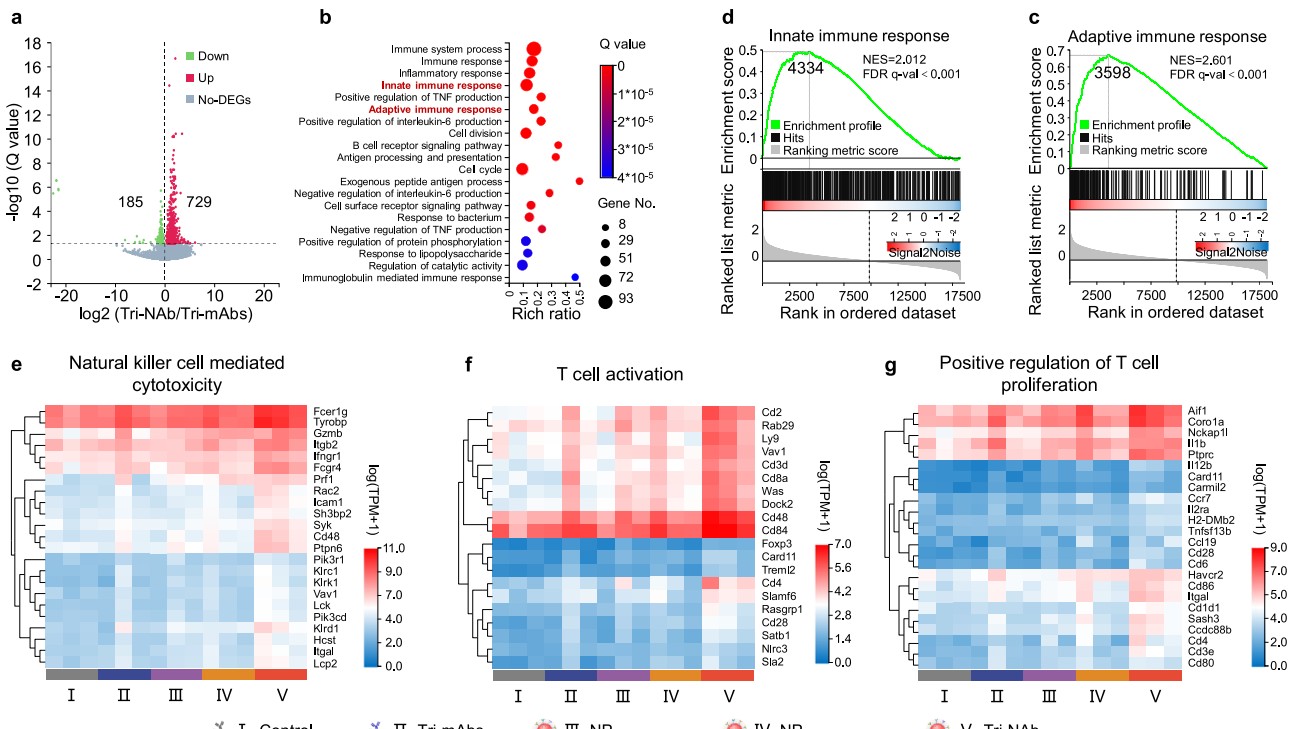

**Fig. 5 | Transcriptomic analysis. a** Volcano plot displaying the distributions of upregulated (729) and downregulated genes (185) (≥2-fold difference, Q value < 0.05) in the Tri-NAb group compared to the Tri-mAbs group. DEGs, differentially expressed genes; non-DEGs, non-differentially expressed genes. **b** Gene Ontology (GO) analysis of biological process annotations of DEGs in (**a**) the Tri-NAb group indicating that the DEGs were mainly involved in immune pathways, including innate and adaptive immune responses. Gene set enrichment analysis (GSEA) of DEGs enriched in innate (**c**) and adaptive immune response pathways (**d**) after treatment with Tri-NAb. The green line indicates the gene enrichment score (ES) change curve. Each black vertical line represents one gene, and genes enriched in either condition are shown at the left (Tri-NAb) or right (Tri-mAbs) parts of the graph. NES, normalized enrichment score. FDR q-val, false discovery rate q-value. Heatmaps of different groups of upregulated DEGs related to NK-cell-mediated cytotoxicity (**e**) T-cell activation (**f**) and positive regulation of T-cell proliferation (**g**) pathways. Source data are provided as a Source Data file.

lower doses, such as 3.75 mg/kg Tri-NAb or 1.25 mg/kg αPDL1/α4-1BB/αNKG2A, Tri-NAb effectively controlled tumor progression (Figs. 6k–m). These findings collectively indicate that by enhancing the infiltration and activation of NK and CD8⁺ T cells, Tri-NAb mediates efficient melanoma regression, suggesting its universal potential as an excellent antitumor therapy.

### Hu Tri-NAb exhibited potent killing activity on patient-derived colon cancer organoids and human HT-29 colorectal cancer in humanized mice

Tumor organoids effectively preserve the biological characteristics and heterogeneity of tumor tissues, making them exemplary models for tumor research. In this study, we generated colon cancer organoids from patient-derived colon cancer tissues to further investigate the ability of Tri-NAb to kill cancer cells. Humanized monoclonal antibodies, namely, anti-human PDL1 antibody (Hu-αPDL1), anti-human 4-1BB antibody (Hu-α4-1BB), and anti-human TIGIT antibody (Hu-αTIGIT), were expressed with exceptional purity, exceeding 95%, and exhibited high affinity toward their respective antigens (Supplementary Figs. 38 and 39). Subsequently, we conjugated anti-human IgG (Fc)-specific antibodies (Hu αFc) onto the surface of APCN to create an APCN-based human mAbs-NA known as Hu APCN@NA. This mixture was then mixed with Hu-αPDL1, Hu-α4-1BB, and Hu-αTIGIT to generate a tri-specific Nano-Antibody called Hu Tri-NAb (Figs. 7a, b and Supplementary Fig. 40). Primary human CD56⁺ NK and CD8⁺ T cells were cocultured with CFSE-labeled colon cancer organoids, and dead cells were stained with propidium iodide (PI) for visualization to assess the killing effect on colon cancer cells (Fig. 7c). The bright field and fluorescence images clearly depicted the hierarchical morphology of the constructed colon cancer organoids (Figs. 7d, e). The proportion of organoid killing (CFSE⁺ PI⁺ cells) mediated by the Hu Tri-NAb was significantly increased, indicating that the Hu Tri-NAb exerted a favorable cytotoxic effect on human tumors in vitro (Figs. 7f, g). Furthermore, the ability of the Hu Tri-NAb to inhibit tumor growth in vivo was evaluated in immunodeficient NCG-hIL15 mice reconstituted with human PBMCs (Fig. 7h). Remarkably, compared with Hu Tri-mAbs treatment alone, Hu Tri-NAb treatment effectively suppressed the growth of HT-29 human colon cancer cells (Figs. 7i–k). ELISAs revealed notably elevated levels of human IFN-γ, granzyme B, and perforin in the group treated with Hu Tri-NAb (Figs. 7l–n). Immunofluorescence staining further indicated enhanced infiltration of human NK and T cells within the tumor tissue of mice treated with Hu Tri-NAb (Fig. 7o). These findings suggest that the superior inhibitory effect of Hu Tri-NAb on human tumors may be attributed to intensified innate and adaptive immune responses involving enhanced cytolytic activity of NK and T cells, highlighting the tremendous potential of Hu Tri-NAb as a therapeutic approach for treating human tumors.

## Discussion

Although the development of immune checkpoint inhibitors represents a revolutionary milestone in the field of immuno-oncology, the identification of potential drug targets remains limited, and immune-related adverse events and drug resistance cannot be disregarded[40,41]. Given these circumstances, researchers have shifted their focus to bi- or multi-specific antibodies capable of recognizing two or more targets[42,43]. Compared to the challenging and costly research and development involving tri-specific antibodies, harnessing nanotechnology to construct "tri-

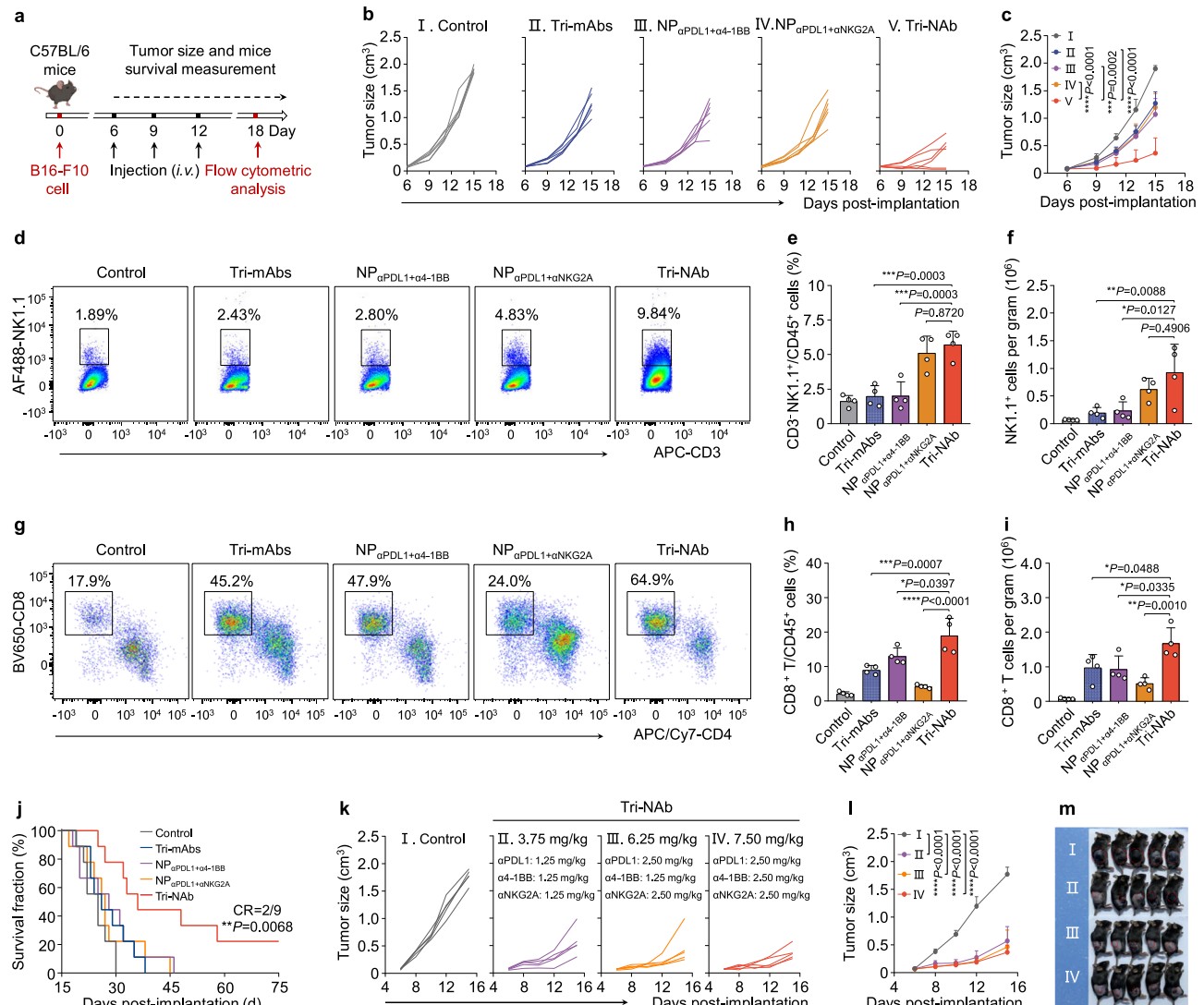

**Fig. 6 | Tri-NAb effectively inhibited the progression of murine melanoma in vivo by simultaneously promoting the infiltration of NK and CD8+ T cells.**
**a** Experimental scheme of the subcutaneous B16-F10 mouse melanoma model in male C57BL/6 mice. Different formulations with equivalent doses of αPDL1, α4-1BB, and αNKG2A (2.5 mg/kg each) were intravenously (i.v.) administered via the tail vein on Days 7, 10, and 13 after B16-F10 tumor cell inoculation. Individual (**b**) and average (**c**) tumor growth kinetics in mice treated with different formulations. The growth curves represent the means ± s.d. (*n* = 6 mice per group). Representative scatter plots of flow cytometry data showing the number of NK cells as a percentage of the CD3⁻ cell population (**d**) or CD8+ T cells as a percentage of the CD3+ cell population (**g**) in the tumor after different treatments, as indicated. Quantitative results for the numbers of NK cells (**e**) or CD8+ T cells (**h**) as a percentage of the total CD45+ cell population in the tumor. Quantitative results for the total numbers of NK cells (**f**) or CD8+ T cells (**i**) per gram of tumor tissue. FACS data are presented as the means ± s.d. (*n* = 4 biologically independent mice). **j** Survival curves of each group

of female C57BL/6 mice with melanoma are presented. MS, median survival time. CR, complete response. Data are presented as means ± s.d. (*n* = 9 mice per group). Individual (**k**) and average (**l**) tumor growth kinetics in female mice treated with different formulations: PBS, Tri-NAb containing equivalent doses of αPDL1, α4-1BB, and αNKG2A (1.25 mg/kg each) at a total dose of 3.75 mg/kg; Tri-NAb containing 2.5 mg/kg of αPDL1, 1.25 mg/kg of α4-1BB, and 2.5 mg/kg of αNKG2A at a total dose of 6.25 mg/kg; and Tri-NAb containing equivalent doses of αPDL1, α4-1BB, and αNKG2A (2.5 mg/kg each) at a total dose of 7.50 mg/kg. The growth curves represent the means ± s.d. (*n* = 5 mice per group). **m** Images of mice showing tumors (red dashed circles) at the treatment endpoint. For (**j**), statistical significance was calculated using the log-rank (Mantel−Cox) test. For the other variables, statistical significance was calculated using one-way ANOVA with Tukey's post hoc test. *P* < 0.05; **P* < 0.01; ***P* < 0.001; and ****P* < 0.0001. Figure 6a was created with BioRender.com under a CC-BY-NC-ND license. Source data are provided as a Source Data file.

specific antibody mimics" for targeted cancer therapy signifies a significant breakthrough and innovative direction. Current strategies for cancer immunotherapy predominantly concentrate on antigen-specific adaptive immunotherapy with minimal simultaneous targeting of NK and CD8+ T cells. This approach represents a fresh opportunity to further enhance antitumor immunity.

The coexpression of a multitude of coinhibitory receptors (also referred to as inhibitory checkpoints, such as PDL1/PD1, CTLA4, NKG2A, and TIGIT) and costimulatory receptors (such as 4-1BB, OX40,

and CD28) by NK and CD8+ T cells concurrently regulates their function, proliferation, and survival[5,44,45]. Accumulating evidence strongly suggests that the activation of costimulatory pathways may enhance the efficacy of checkpoint inhibition and result in enduring antitumor responses[46-49]. In this study, we initially developed a tri-specific Nano-Antibody (Tri-NAb) that targets PDL1/4-1BB/NKG2A to simultaneously activate NK and CD8+ T cells, thereby harnessing the innate and adaptive immune systems of the host for effective cancer eradication. We demonstrated that the Tri-NAb, by simultaneously engaging both

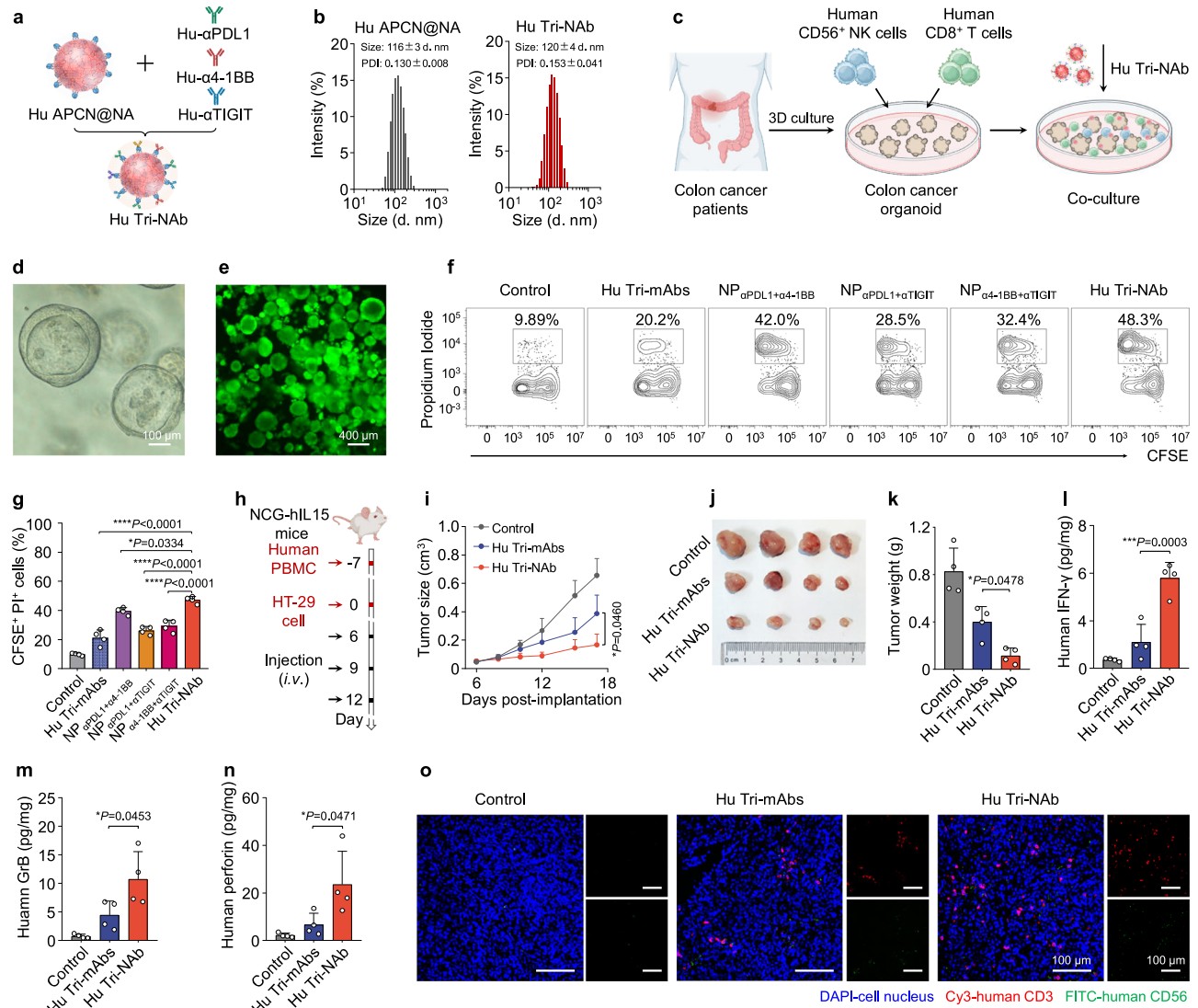

**Fig. 7 | Hu Tri-NAb exhibited potent killing activity on patient-derived colon cancer organoids and human HT-29 colorectal cancer in humanized mice. a** Scheme of the construction of Hu Tri-NAb through gentle mixing of Hu APCN@NA and Hu-αPDL1, Hu-α4-1BB, and Hu-αTIGIT. Similar to Fig. 2i, Hu APCN@NA was prepared by coupling Hu αFc on the surface of APCN. **b** Size distribution of Hu APCN@NAs and Hu Tri-NAb determined using DLS. **c** An experimental scheme illustrating the study of the cytotoxicity of human primary CD56+ NK and CD8+ T cells from human peripheral blood mononuclear cells (PBMCs) toward colon cancer organoids from patient-derived colon cancer tissues after an incubation with 1) control, 2) Hu Tri-mAbs (Hu-αPDL1 & Hu-α4-1BB & Hu-αTIGIT), 3) NP αPDL1+α4-1BB, 4) NP αPDL1+αTIGIT, 5) NP α4-1BB+αTIGIT, or 6) Hu Tri-NAb. The organoids were first labeled with CFSE, and all the cells were stained with propidium iodide (PI). The proportion of CFSE+ PI+ cells was detected using flow cytometry. **d** Brightfield image of colon cancer organoids. **e** Fluorescence image of CFSE-labeled colon cancer organoids. **f** Representative scatter plots of flow cytometry data and corresponding quantification results (**g**) showing the number of CFSE+ PI+ cells as a

percentage of the total cells. Data are presented as means ± s.d. (n = 4 biologically independent samples). **h** Experimental scheme of the subcutaneous HT-29 human colon xenograft tumor model in female NCG-hIL15 mice. **i** Average tumor growth kinetics in mice treated with different formulations. Data are presented as means ± s.d. (n = 4 mice per group). **j** Tumor images collected from treatment endpoints. **k** Tumor weights after different treatments. Data are presented as means ± s.d. (n = 4 mice per group). The levels of human IFN-γ (**l**), human granzyme B (**m**), and human perforin (**n**) in treated HT-29 tumors were examined using ELISAs. Data are presented as the means ± s.d. (n = 4 biologically independent samples). **o** Representative fluorescence images of tumor lesions after staining with DAPI (blue), anti-human CD56 (FITC, green), and anti-human CD3 antibodies (Cy3, red). Statistical significance was determined using one-way ANOVA with Tukey's post hoc test. *P < 0.05; **P < 0.01; ***P < 0.001; and ****P < 0.0001. Figures 7c and h were created with BioRender.com under a CC-BY-NC-ND license. Source data are provided as a Source Data file.

costimulatory and coinhibitory receptors, can more efficiently promote the proliferation and activation of NK/CD8+ T cells than can bispecific NAbs, which modulate only one immunomodulatory target. Importantly, Tri-NAb exhibits high-affinity binding to tumor cells, as well as NK/CD8+ T cells, through antigen–antibody recognition mechanisms, thereby substantially augmenting cell interactions while enhancing granzyme B and perforin release by NK/CD8+ T cells for the potent killing of neighboring tumor cells. Our comprehensive studies confirmed that Tri-NAb reinforced both innate and adaptive immune

responses at relatively low doses with reduced administration frequency compared to previously reported treatments involving mAbs targeting PDL1, 4-1BB, or NKG2A[9,50,51]. Importantly, our injectable doses of Tri-NAb exhibited good safety profiles. Moreover, our constructed humanized Tri-NAb (Hu Tri-NAb) targeting human PDL1, 4-1BB, and TIGIT displayed remarkable efficacy against patient-derived colon cancer organoids and effectively suppressed HT-29 human colon cancer growth in humanized mice. Overall, our research confirms the feasibility and effectiveness of an innovative tri-specific Nano-

Antibody that simultaneously targets NK and CD8+ T cells, with tremendous potential for clinical applications. Additionally, our platform offers flexibility for replacing therapeutic targets or combinations, making it applicable to diseases other than cancer. However, further validation studies are necessary, including toxicity assessments in large primates and comprehensive comparisons with other recently developed bi- or tri-specific antibodies[52].

## Methods

### Ethical statement

This research complies with all relevant ethical regulations. All the experiments in this research were approved by the Animal Care and Use Committee at South China University of Technology (SCUT) (official approval number: 2019012). According the guidelines of the Institutional Animal Care and Use Committee (IACUC) and the local authorities (the Animal Care and Use Committee at SCUT), the tumors in mice are allowed to grow to 2 cm$^3$ as long as the mouse remains otherwise healthy. In some cases, this limit has been exceeded the last day of measurement and the mice were immediately euthanized.

### Materials and reagents

Human serum albumin (HSA) was purchased from Equitech Bio, Inc. Stearic acid, palmitic acid, and sodium periodate ($NaIO_4$) were purchased from Shanghai Aladdin Bio-Chem Technology Co., Ltd. Sodium borohydride ($NaBH_4$) was purchased from Energy Chemicals. Various polyesters with different weight-average molecular weights ($\overline{M_w}$), including poly(L-lactide) (PLLA-COOR, R = -$(CH_2)_{11}CH_3$, $\overline{M_w}$: 7.2, 36, 137, and 240 K; PLLA-COOH, $\overline{M_w}$: 1.5, 17, 40, 66, 160, and 210 K); poly(D,L-lactide) (PDLLA-COOH, $\overline{M_w}$: 6.0, 20, 54, 115, and 220 K); and poly(lactic-co-glycolic acid) (PLGA75/25-COOH, $\overline{M_w}$: 6.2, 12, 24, 53, 75, 105, and 158 K), were purchased from Jinan Daigang Biomaterial Co., Ltd. Goat anti-rat IgG (Fc-specific) (αFc) and rabbit anti-human IgG (Fc-specific) antibodies (Hu αFc) were obtained from Rockland Immunochemicals, Inc. *In Vivo*MAb anti-mouse NKG2A/C/E antibody (clone: 20D5), anti-mouse PDL1 antibody (clone: 10 F.9G2), anti-mouse 4-1BB antibody (clone: 3H3), rat IgG2a isotype control (anti-trinitrophenol, clone: 2A3), and rat IgG2b isotype control (anti-keyhole limpet hemocyanin, clone: LTF-2) were obtained from BioX Cell. Recombinant mouse PDL1, 4-1BB, and NKG2A proteins (rm PDL1, rm 4-1BB, and rmNKG2A) were purchased from Sino Biological, Inc. Fluorochrome-labeled antibodies for flow cytometry were purchased from BioLegend. Anti-rat IgG (whole molecule)-gold antibody, collagenase type IV, and the PKH26 Red Fluorescent Cell Linker Mini Kit were obtained from Sigma–Aldrich. The CellTrace™ CFSE Cell Proliferation Kit, Alexa Fluor™ 488 NHS ester (AF488), Alexa Fluor™ 555 NHS ester (AF555), and Alexa Fluor™ 647 NHS ester (AF647) were purchased from Thermo Fisher Scientific. iFluor® 488 succinimidyl ester (iFluor 488) was obtained from AAT Bioquest. Bovine serum albumin (BSA), hyaluronidase and deoxyribonuclease I were purchased from Sangon Biotech (Shanghai) Co., China. A cell plasma membrane staining kit with DiD was purchased from Beyotime Biotechnology Co., Ltd. Sulfocyanine 5 NHS ester (Cy5), fluorescein isothiocyanate (FITC), and rhodamine B (RhoB) were obtained from J&K Scientific. Cell staining and antibody labeling were performed according to the manufacturer's instructions.

### Cell culture and animals

B16-F10 murine melanoma and MC38 murine colon cancer cells were obtained from the American Type Culture Collection (ATCC). MC38 cells expressing firefly luciferase (MC38-luc) were generated by transfecting luc-encoding lentiviral vectors (Vectorbuilder) into MC38 cells. The cells were maintained in Dulbecco's modified Eagle's medium supplemented with 10% fetal bovine serum (FBS, Gibco) and 1% penicillin/streptomycin (Invitrogen) in a humidified atmosphere containing 5% carbon dioxide at 37 °C. Primary CD8+ T cells were isolated

from the spleens of C57BL/6 or OT-I transgenic mice using mouse CD8a (Ly2) MicroBeads (Miltenyi Biotec) and then cultured in RPMI-1640 medium supplemented with 10% fetal bovine serum (FBS), 1% penicillin/streptomycin, 1% GlutaMAX (Life Technologies), 10 mM HEPES (Life Technologies), 1.0 mM sodium pyruvate (Life Technologies), 55 μM 2-mercaptoethanol (Life Technologies), and 10 ng/mL IL-2 (PeproTech). CD49b+ NK cells were isolated using mouse CD49b (DX5) MicroBeads (Miltenyi Biotec) and cultured in RPMI-1640 medium supplemented with 10% FBS, 1% penicillin/streptomycin, 1% GlutaMAX, 10 mM HEPES, 1.0 mM sodium pyruvate, 55 μM 2-mercaptoethanol, 10 ng/mL IL-2, and 20 ng/mL IL-15 (PeproTech). Hoechst DNA staining and agar culture were used to confirm the lack of *Mycoplasma* staining in all cells.

Female/male C57BL/6 (hnslkjd005) and female ICR mice (hnslkjd003) were purchased from Hunan Silaike Jingda Laboratory Animal Technology Co., Ltd (Production license number: SCXK (Xiang) 2021-0002/SCXK (Xiang) 2024-0009, Changsha, China). Female NOD/ShiLtJGpt-Prkdc$^{em26Cd52}$Il2rg$^{em26Cd22}$Il15$^{em1Cin(hIL15)}$/Gpt mice (NCG-hIL15, Strain NO. T004886) were purchased from GemPharmatech (Production license number: SCXK (Su) 2023-0009, Nanjing, China). All the mice were maintained in a specific pathogen-free (SPF) environment with controlled temperature (~22 °C) and humidity (50 ± 15%) under 12 h light/dark cycle at the South China University of Technology (SCUT) Animal Facility. Mice aged 6–8 weeks were used for experiments. All animal experiments were approved by the Animal Care and Use Committee of the SCUT, and every effort was made to minimize the pain caused by the experiments.

### Preparation and characterization of albumin/polyester composite nanoparticles (APCNs)

APCNs were formulated using a single emulsification-solvent evaporation method. Briefly, aqueous solutions of BSA or HSA (10 mg/mL), LCFAs (long-chain fatty acids), or polyesters dissolved in chloroform (5 mg/mL) were ultrasonically mixed at a volume ratio of 5/1 in an ice bath using an ultrasonic processor (Sonics & Materials, Inc.) to form an emulsion. The organic solvents were demulsified and removed using a rotary evaporator (IKA). Stearic or palmitic acid was first selected to prepare NPs with LCFA/BSA weight ratios of 1/10, 2/10, and 4/10. Various polyesters with different $\overline{M_w}$s, including PLLA-COOR (R = -$(CH_2)_{11}CH_3$, $\overline{M_w}$:7.2, 36, 137, or 240 K), PLLA-COOH ($\overline{M_w}$:1.5, 17, 40, 66, 160, or 210 K) and PDLLA-COOH ($\overline{M_w}$:6.0, 20, 54, 115, or 220 K) and PLGA75/25-COOH ($\overline{M_w}$:6.2, 12, 24, 53, 75, 105, or 158 K), were utilized to prepare NPs by setting the weight ratio of polyester to BSA (P/B) to 1/5, 1/7.5, and 1/10 to optimize the formulation, as described in the methods above. The hydrodynamic size, polydispersity index (PDI), and zeta potential of the different nanoparticles were measured using a Zetasizer Nano ZSE instrument (Malvern Panalytical). The albumin loading efficiency was quantified using HPLC (Waters, Ultrahydrogel Column 500 Å). The Kjeldahl method was used to determine the total protein content. The purified APCNs were obtained by high-speed centrifugation at 4 °C (21000 g for 60 min). The morphology of the APCNs was tested using scanning electron microscopy (SEM, Merlin, Zeiss) and transmission electron microscopy (TEM, Talos F200x, Thermo Fisher). Elemental mapping (e.g., C/H/O/N/S) was performed using energy dispersive spectroscopy (EDS, Thermo Fisher Scientific). RhoB (rhodamine B) was coupled to PLLA via a chemical reaction, and RhoB-labeled PLLA was obtained using the precipitation method. iF488-labeled HSA was obtained according to the manufacturer's instructions. Fluorescent dye-labeled APCNs were further prepared using the aforementioned methods and then immersed in stochastic optical reconstruction microscopy (STORM) imaging buffer (Smart Buffer Kit, Abbelight). STORM image acquisition was performed using an Abbelight superresolution microscope (Abbelight) equipped with two Oxxius lasers at 488 and 532 nm (Oxxius) and NEO software (Abbelight). The hydrodynamic size and PDI of APCNs in ddH$_2$O, PBS,

and PBS supplemented with 10% FBS at 37 °C for 8 days were measured using a Zetasizer Nano ZSE instrument (Malvern Panalytical) to evaluate the stability in vitro.

## Molecular dynamics simulation

Molecular dynamics (MD) simulations were performed by the Fuda Testing Group using the GROMACS 2019.6 program under constant temperature, constant pressure, and periodic boundary conditions. The General AMBER Force Field (GAFF) was applied to the polymer molecule simulation, while the Amber14SB all-atom force field and TIP3P water model were used for protein simulations. A leapfrog integrator was used, and the integral time step was set to 2 fs. All bonds involving hydrogen atoms were constrained using the linear constraint solver (LINCS) algorithm during the MD simulation. The electrostatic interactions were calculated using the particle mesh Ewald (PME) algorithm. The cutoff for the nonbonded interactions was 10 Å. The system was then controlled at 298 K using the V-rescale temperature coupling scheme and at 1 bar using the Parrinello–Rahman pressure coupling scheme. The steepest descent method was first used to minimize the energy of the two systems to eliminate the close contact between the atoms. Constant NVT MD simulations were then conducted at 298 K with a time step of 1 ns, followed by another 1 ns NPT simulation. Finally, a 10 ns simulation was conducted on the system, and the trajectory was saved every 10 ps. The GROMACS embedded program and VMD were used to visualize the simulation results.

## Calculations of the binding free energy

The binding free energy (ΔGbind) between PLLA and HSA was calculated using a previously reported molecular mechanics/Poisson–Boltzmann surface area (MM/PBSA) approach. ΔGbind contains the following three energy terms:

$$\Delta Gbind = \Delta Ggas + \Delta Gsolv - T\Delta S \qquad (1)$$

$$\Delta Ggas = \Delta Ebond + \Delta Eelec + \Delta EvdW \qquad (2)$$

$$\Delta Gsolv = \Delta Gelec, solv + \Delta Gnp, solv \qquad (3)$$

where Δ Ggas is the energy of the gas phase, including the internal bonding energy (ΔEbond), electrostatic energy (ΔEelec), and van der Waals (ΔEvdW) energy; ΔGsolv is the solvation free energy, including the electrostatic solvation energy (ΔGelec, solv) approximated using the Poisson–Boltzmann model and the nonpolar solvation energy (ΔGnp, solv) calculated using the following equation:

$$\Delta Gnp, sol = \gamma SASA + b \qquad (4)$$

where γ is the surface tension proportionality constant (2.27 kJ/(mol·nm$^2$)) and b is a constant (b = 3.85 kJ/mol). SASA is defined as the solvent-accessible surface area, which is calculated using a linear combination of the pairwise overlap (LCPO) method. The negative energy values suggested a contribution to the stabilization of the PLLA-HSA complex.

## Preparation and characterization of APCN@NA

The carbohydrate residues on the Fc portion of αFc were first oxidized by 0.01 M NaIO$_4$ in 0.05 M acetate buffer (pH 4.2) for 2 h at 4 °C. Subsequently, αFc was immobilized on the APCN surface through the reaction of the aldehyde and amino groups on the albumins. The pH was adjusted to 9.0 to improve the reaction efficiency. Finally, NaBH$_4$ was added every 2 h to reduce the Schiff base intermediates and generate stable covalent linkages between APCN and αFc. Unbound αFc in the supernatant was examined using ELISA (goat IgG ELISA Kit,

Alpha Diagnostic International). The following formula was used to quantify the binding efficacy (BE) of αFc: BE = (A-B)/A, where A is the feed content of αFc and B is the αFc content in the supernatant. The hydrodynamic size, PDI, and zeta potential of APCN@NA were characterized using a Zetasizer Nano ZSE instrument (Malvern Panalytical) as described previously. The morphology of APCN@NA was imaged using SEM. The hydrodynamic size and PDI of APCN@NA in ddH$_2$O, PBS, and serum (PBS containing 10% FBS) buffer were monitored at 37 °C for 8 days to evaluate its in vitro stability. MST (Monolith NT.115, NanoTemper Technologies) was used to test the affinity of APCN@NA for different subtypes of antibodies (rat IgG2a/rat IgG2b), and rat IgG2a and IgG2b were labeled with AF488 according to the manufacturer's instructions. Different concentrations of APCN@NA were added to 0.015 μM AF488-labeled rat IgG2a or IgG2b to detect the interactions.

## Construction and characterization of a tri-specific Nano-Antibody (Tri-NAb)

NP$_{\alpha PDL1+\alpha 4-1BB}$ was prepared by gently mixing αPDL1 and α4-1BB with APCN@NA at an αPDL1/α4-1BB ratio of 1:1 (w/w) and incubated at 4 °C for 24 h. NP$_{\alpha PDL1+\alpha NKG2A}$ was prepared by gently mixing αPDL1 and αNKG2A with APCN@NA at an αPDL1/αNKG2A ratio of 1:1 (w/w) and incubated at 4 °C for 24 h. The Tri-NAb was prepared by gently mixing αPDL1, α4-1BB, and αNKG2A with APCN@NA at an αPDL1/α4-1BB/αNKG2A ratio of 1:1:1 (w/w) and incubated at 4 °C for 24 h. NP$_{\alpha PDL1+\alpha 4-1BB}$, NP$_{\alpha PDL1+\alpha NKG2A}$, and Tri-NAb were further characterized using SEM. The mixture was purified via high-speed centrifugation (21000 g for 60 min) at 4 °C, and the unbound αPDL1, α4-1BB, and αNKG2A in the supernatant were tested via ELISA to quantify the binding efficiency. Briefly, ELISA plates (Corning) were coated with 100 μL (5 μg/mL) of rmPDL1, rm4-1BB, or rmNKG2A overnight at 4 °C, blocked with 2% BSA for 1 h at room temperature (RT) and then incubated for 1 h at RT with the collected supernatant or free mAbs (αPDL1, α4-1BB, or αNKG2A) used as standard samples. After washing, 100 μL (1 μg/mL) of HRP-conjugated goat anti-rat IgG antibody (Sino Biological, Inc.) was added as the detection antibody and incubated for 1 h at RT, followed by the addition of 100 μL of 3,3′,5,5′-tetramethylbenzidine (TMB, Abcam). After another 10 min of incubation, the stop solution was added, and the absorption value at 450 nm was detected using an Infinite® 200 PRO microplate reader (Tecan). AF488-labeled α4-1BB, AF555-labeled αNKG2A, and AF647-labeled αPDL1 were mixed with APCN@NA at an αPDL1/α4-1BB/αNKG2A ratio of 1:1:1 (w/w) and incubated at 4 °C for 24 h to further confirm the successful conjugation of the three antibodies. The fluorescent dye-conjugated Tri-NAb was purified by high-speed centrifugation. The three types of fluorescence were monitored using an HPLC instrument equipped with a fluorescence detector (Waters).

## Antigen-binding capability

ELISA plates (Corning) were coated with 100 μL (5 μg/mL) of rmPDL1, rm4-1BB or NKG2A for 2 h at 37 °C, followed by blocking with 2% BSA in PBST (PBS containing 0.1% Tween® 20 detergent) for 1 h at RT (room temperature). A series of concentrations of free mAb (αPDL1, α4-1BB, or αNKG2A) or Tri-NAb were added, and the mixture was incubated for 1 h at RT. The attached antibodies were examined as described above. The dissociation constant, Kd, was obtained by plotting the normalized absorption values versus the concentrations of αPDL1, α4-1BB, or αNKG2A using PRISM software (GraphPad).

## The binding of Tri-NAb to tumor, NK, or CD8$^+$ T cells in vitro

B16-F10 tumor cells were stimulated with IFN-γ (20 ng/ml, Peprotech) for 24 h. Murine spleens were carefully removed from female C57BL/6 mice and washed three times with precooled sterile PBS to obtain primary mouse NK and CD8$^+$ T cells. The spleens were placed on steel

wire mesh in precooled sterile PBS for gentle fragmentation, and the splenocyte suspension was filtered through 40-μm nylon mesh. Red blood cells were removed using erythrocyte lysis buffer (Biosharp), splenocytes were resuspended in magnetic-activated cell sorting (MACS) buffer, and NK and CD8+ T cells were isolated using mouse CD49b (DX5) and CD8a (Ly2) MicroBeads (Miltenyi Biotec). For NK-cell stimulation, isolated NK cells were incubated with IL-2 (20 ng/ml, PeproTech) and IL-15 (20 ng/ml, PeproTech) for 4 days. For T-cell stimulation, isolated CD8+ T cells were incubated with plate-bound anti-CD3 antibodies (5 μg/mL, BioLegend) and soluble anti-CD28 antibodies (5 μg/mL, BioLegend) for 24 h. Stimulated B16-F10 cells ($1.0 \times 10^4$/well), NK cells ($5.0 \times 10^4$/well), or CD8+ T cells ($5.0 \times 10^4$/well) were seeded in 96-well plates with an ultralow attachment surface (Corning). PBS or FITC-labeled Tri-NAb was added and coincubated for 6 h, and 0.4% trypan blue was added to another group treated with FITC-labeled Tri-NAb to quench the extracellular fluorescence. The fluorescence intensities of tumor, NK, or CD8+ T cells were analyzed using a BD FACSCelesta™ flow cytometer (BD Biosciences). Data were analyzed using FlowJo software (version 10.0.7, TreeStar Inc., Ashland, USA).

### Proliferation of Tri-NAb-mediated NK and CD8+ T cells
Mouse NK cells ($1.0 \times 10^5$/well) were labeled with carboxyfluorescein succinimidyl ester (CFSE), seeded in 96-well plates, stimulated with IL-2/IL-15 (20 ng/ml) and then treated with PBS, α4-1BB, $NP_{α4-1BB}$, $NP_{αPDL1+α4-1BB}$, $NP_{αPDL1+α4-1BB}$ & $NP_{αPDL1+αNKG2A}$ (mixed NPs) or Tri-NAb for 3 days. The doses of α4-1BB, αPDL1, and αNKG2A were set to 5 μg/mL. Similarly, mouse CD8+ T cells ($1.0 \times 10^5$/well) were labeled with CFSE, seeded in 96-well plates precoated with an anti-CD3 antibody (0.01 μg/mL), and treated as described above. Unstimulated CFSE-labeled NK or CD8+ T cells were used as negative controls. The proportion of DAPI⁻CFSE^low cells was analyzed using a FACSCelesta flow cytometer (BD Biosciences). The data were analyzed using FlowJo software.

### Simultaneous bridging of NK and CD8+ T cells to tumor cells in vitro
MC38 cells, isolated NK cells, and CD8+ T cells were stimulated as described above to visualize Tri-NAb-mediated interactions between immune and tumor cells. Specifically, MC38 cells were labeled with PKH26 dye, whereas NK and CD8+ T cells were labeled with CFSE and DiD dye, respectively. PKH26-labeled MC38 cells ($2.5 \times 10^4$) were seeded in glass-bottomed culture dishes (Wuxi NEST Biotechnology Co., Ltd.) and stimulated with IFN-γ (20 ng/ml, PeproTech) for 24 h. After the culture medium was removed, CFSE-labeled CD8+ T cells ($2.5 \times 10^5$) or DiD-labeled NK cells ($2.5 \times 10^5$) were added, and then the cocultured cells were treated with IgG, Tri-mAbs, $NP_{αPDL1+α4-1BB}$, $NP_{αPDL1+αNKG2A}$, $NP_{α4-1BB+αNKG2A}$ or Tri-NAb; the concentrations of αPDL1, α4-1BB, or αNKG2A were 10 μg/mL, and the concentration of IgG was 30 μg/mL. After a coincubation for 6 h, the supernatant containing unbound NK and CD8+ T cells was removed, and adherent cells in the dishes were washed three times, fixed, and imaged using a Zeiss LSM880 laser scanning confocal microscope.

### Measurement of mouse IFN-γ, granzyme B, and perforin 1 levels via ELISA
MC38 cells ($5.0 \times 10^3$) were seeded in 96-well plates (Costar) and stimulated with IFN-γ (20 ng/ml; PeproTech) for 24 h. The culture medium was then removed, and NK ($5.0 \times 10^4$/well) and CD8+ T cells ($5.0 \times 10^4$/well) were added. Cocultured cells were treated with IgG, Tri-mAbs, $NP_{αPDL1+α4-1BB}$, $NP_{αPDL1+αNKG2A}$, or Tri-NAb; the concentrations of αPDL1, α4-1BB, or αNKG2A were 10 μg/mL, and the concentration of IgG was 30 μg/mL. After coincubation for 48 h, IFN-γ, granzyme B, and perforin 1 in the collected cell supernatant were quantified via a mouse IFN-γ ELISA kit (Dakewe Biotech), mouse granzyme B ELISA kit

(Abcam) and mouse perforin 1 ELISA kit (Abbexa) according to the manufacturer's protocol.

### Detection of both NK- and CD8+ T-cell-mediated cytotoxicity in vitro
MC38-luc cells ($5.0 \times 10^3$) were seeded in 96-well plates (Costar) and stimulated with IFN-γ (20 ng/ml; Peprotech) for 24 h. The culture medium was then removed, and stimulated NK ($5.0 \times 10^4$/well) and CD8+ T cells ($5.0 \times 10^4$/well) were added. Cocultured cells were treated with IgG, Tri-mAbs, $NP_{αPDL1+α4-1BB}$, $NP_{αPDL1+αNKG2A}$, or Tri-NAb; the concentrations of αPDL1, α4-1BB, or αNKG2A were 10 μg/mL, and the concentration of IgG was 30 μg/mL. After coincubation for 48 h, the cell culture supernatant was discarded, and 100 μL of a D-Luciferin potassium salt solution (150 μg/mL) was added to the reaction mixture, which was incubated for 5 min to allow it to enter the cells. Luminescence intensity was analyzed immediately using an Infinite® 200 PRO microplate reader (Tecan).

### Detection of fratricide in NK and CD8+ T cells cocultured with Tri-NAb
CFSE-labeled NK or CD8+ T cells (green) were treated with Tri-NAb (10 μg/mL per mAb), and cell behavior was evaluated within 6 h using the ImageXpress® Micro Confocal High-Content Imaging System (Molecular Devices) to observe whether NK or CD8+ T cells were harmed. Propidium iodide dye (50 μg/mL) was added before imaging to distinguish between living and dead cells (red). An image of the NK cells is shown in Supplementary Video 1. An image of CD8+ T cells is shown in Supplementary Video 2. Coincubated DiD-labeled NK (blue) and CFSE-labeled CD8+ T cells (green) were treated with Tri-NAb (10 μg/mL per mAb) using the same procedure described above to observe whether binding led to effector cell fratricide. The animation is shown in Supplementary Video 3.

### Evaluation of pharmacokinetics and biodistribution in vivo
For in vivo pharmacokinetics, αPDL1 was first labeled with Cy5 and then conjugated to APCN@NA ($NP_{αPDL1}$). A solution of free αPDL1 or $NP_{αPDL1}$ was intravenously injected into female C57BL/6 mice at equivalent αPDL1 doses (2 mg/kg), and a 100 μL blood sample was collected from the mice at different time points using an anticoagulation tube. The fluorescence intensity was detected using an Infinite® 200 PRO microplate reader (Tecan).

For the analysis of the in vivo biodistribution, MC38 tumor-bearing female C57BL/6 mice were intravenously treated with Tri-mAbs or Tri-NAb (doses of 2.5 mg/kg of αPDL1, α4-1BB, or αNKG2A). Each mAb was labeled with Cy5 dye. Twelve and 36 h after injection, the major organs and tumor tissues were collected, and the Cy5 signal intensity was monitored using an imaging system (PerkinElmer).

### Evaluation of therapeutic efficacy in vivo
For tumor cell inoculation, $1.0 \times 10^6$ MC38 cells were subcutaneously injected into the right flank of female C57BL/6 mice. A total of $2.5 \times 10^5$ B16-F10 cells were subcutaneously injected into the right flank of male C57BL/6 mice. The major (L) and minor (W) axes of the tumors were measured every two days using Vernier calipers, and the tumor volume (V) was calculated according to the following formula: $V = (L \times W \times W)/2$. Body weight was monitored every two or three days. When a palpable tumor was present, the mice were randomly split into five groups and treated with the corresponding formulations via intravenous injection every three days for a total of three injections (q3dx3) of IgG, Tri-mAbs, $NP_{αPDL1+α4-1BB}$, $NP_{αPDL1+αNKG2A}$, or Tri-NAb (doses of 2.5 mg/kg of αPDL1, α4-1BB, or αNKG2A). For animal welfare, the mice were sacrificed when the tumors reached a volume of 2000 mm³.

The therapeutic effect of the formulations on the subcutaneous MC38-luc tumor model with an initial volume of 150 or 300 mm³ was visualized by subcutaneously injecting $1.0 \times 10^6$ MC38-luc cells into the

right flank of the female C57BL/6 mice. When the tumor size increased to 150 mm$^3$, the mice were randomly divided into three groups and treated with the corresponding formulations on a schedule of intravenous injections every three days for a total of three injections (q3d x 3); animals received the IgG control, Tri-mAbs, or Tri-NAb (doses of 2.5 mg/kg of αPDL1, α4-1BB, or αNKG2A), and tumor changes were observed using IVIS during treatment. A model with larger tumors was constructed by subcutaneously injecting $2.0 \times 10^6$ MC38-luc cells into the right flanks of the mice. When the tumors had grown to 300 mm$^3$, the mice were randomly divided into three groups and treated using the same formula and schedule as described above.

The therapeutic effects of lower doses were further explored by subcutaneously injecting $5.0 \times 10^5$ B16-F10 tumor cells injected into the right flanks of the female C57BL/6 mice. When a palpable tumor was present, the mice were randomly divided into four groups. Then, the mice were treated with the corresponding formulations on a schedule of intravenous injection every 3 days for a total of three injections (q3dx3); mice received the IgG control, 3.75 mg/kg Tri-NAb (doses of 2.5 mg/kg of αPDL1, α4-1BB or αNKG2A), 6.25 mg/kg Tri-NAb (doses of 2.5, 1.25, and 2.5 mg/kg of αPDL1, α4-1BB or αNKG2A, respectively) or 7.5 mg/kg Tri-NAb (doses of 2.5 mg/kg of αPDL1, α4-1BB, or αNKG2A). The tumor volumes and body weights of the mice were monitored every two days. At the end of treatment, the tumors were extracted and photographed.

### Flow cytometry analysis of tumor-infiltrating lymphocytes (TILs) in mice

Male C57BL/6 Mice bearing B16-F10 tumors were treated with IgG, Tri-mAbs, NP$_{αPDL1+α4-1BB}$, NP$_{αPDL1+αNKG2A}$, NP$_{α4-1BB+αNKG2A}$, or Tri-NAb (doses of 2.5 mg/kg of αPDL1, α4-1BB or αNKG2A). The tumors were collected and weighed two days after the last injection, cut into small pieces of the specified weight, and digested using a cocktail of enzymes containing collagenase type IV (1 mg/mL), hyaluronidase (100 μg/mL) and DNase I (100 μg/mL) at 37 °C for 30 min. The digested cells were passed through a 40 μm nylon mesh, and red blood cells were lysed. After counting, 100 μL of the cell suspension ($2.0 \times 10^7$ cells/mL) was obtained for subsequent antibody staining to determine the numbers of live cells, CD45$^+$ immune cells, CD3$^+$ T lymphocytes, CD4$^+$ T cells, CD8$^+$ T cells, and NK1.1$^+$ cells in the tumors using a BD FACSCelesta™ flow cytometer (BD Biosciences). All surface antibodies were used at 1:100 dilution. The data were analyzed using FlowJo software (version 10.0.7, TreeStar Inc., Ashland, USA).

### Transcriptomic analysis

MC38 tumor-bearing female C57BL/6 mice were intravenously treated with IgG, Tri-mAbs, NP$_{αPDL1+α4-1BB}$, NP$_{αPDL1+αNKG2A}$, or Tri-NAb as described above. Two days after the last treatment, the tumor tissues were collected and shipped to the Beijing Genomics Institute (BGI Genomics Co., Ltd.) for analysis. Sequence reads were obtained using the DNBSEQ platform in paired-end mode (150 bp). The sequencing data were checked and filtered using SOAPnuke software (https://github.com/BGI-flexlab/SOAPnuke, version 1.5.2) to obtain clean reads. Clean reads were aligned to the mouse reference genome version GCF_000001635.26_GRCm38.p6 sourced from NCBI using HISAT software (http://www.ccb.jhu.edu/software/hisat, version 2.0.4) and to the mouse reference gene sequence using Bowtie2 software (http://bowtie-bio.sourceforge.net/bowtie2/index.shtml, version 2.2.5). RSEM software (http://deweylab.biostat.wisc.edu/rsem/rsem-calculate-expression.html, version 1.2.8) was used to calculate the gene expression levels of each sample.

### In vivo biological safety evaluation

Female ICR mice (6–8 weeks) were intravenously injected with PBS, APCN, Tri-mAbs, or Tri-NAb (doses of 2.5 mg/kg of αPDL1, α4-1BB or αNKG2A, where the dose of each mAb was the same as that used in the

efficacy study for effective detection) every three days for three repeats. One month after the last injection, peripheral blood and the main organs were collected from the treated mice for serum biochemical and histological analyses. Three months after treatment, the peripheral blood and main organs were also collected from tumor-free mice, and healthy mice from the same batch were used as controls. Multiple indices of liver, renal, and cardiac function were measured using an automatic analyzer (HITACHI). The heart, liver, spleen, lungs, and kidneys were subjected to histological analysis. Images of hematoxylin and eosin (H&E) staining were obtained using a digital pathology scanning system (P250 FLASH, 3D Histech).

### Expression of human monoclonal antibodies

The variable region sequences of the human anti-PDL1, anti-4-1BB, and anti-TIGIT antibodies were derived from atezolizumab (Genentech), utomilumab (Pfizer), and tiragolumab (Roche), respectively, and the antibody constant region sequences were selected from human IgG1, all of which were obtained from online databases. For expression in mammalian cell lines, antibody constructs were first cloned and inserted into pKS expression vectors, and antibody plasmids were constructed by companies offering gene synthesis services. The plasmids were amplified in *Escherichia coli* and transfected into human embryonic kidney (HEK) 293 F cells for transient expression. Cell supernatants were harvested to purify recombinant proteins using Ni-chelating affinity chromatography. The purity was greater than 95%, as determined by SDS–PAGE and HPLC analyses. The antigen affinity was characterized using ELISA.

### Culture and viability of colon cancer organoids in the presence of human immune cells

Colon cancer tissues from colon cancer patients were washed and digested repeatedly and then cultured with Matrigel (Corning), DMEM/F-12 (Dulbecco's modified Eagle's medium/Nutrient Mixture F-12) medium, and IntestiCult™ Organoid Growth Medium (Human). Organoid morphology was photographed in a bright field using a fluorescence microscope (Nikon). First, colon cancer organoids were stimulated with human IFNγ for three days, primary human NK cells (Oricells) were activated with human IL-2 and IL-15 for four days, and primary human CD8$^+$ T cells (Oricells) were stimulated with an anti-CD3 antibody for three days and cultured with human IL-2. Organoids were labeled with CFSE and seeded into ultralow-attachment multiple-well plates (Corning). Stimulated human NK and CD8$^+$ T cells were added, and then cocultured cells were treated with IgG, Hu Tri-mAbs, NP$_{αPDL1+α4-1BB}$, NP$_{αPDL1+αTIGIT}$, NP$_{α4-1BB+αTIGIT}$, or Tri-NAb; the concentrations of αPDL1, α4-1BB, or αTIGIT were 10 μg/mL, and the concentration of IgG was 30 μg/mL. For Hu Tri-mAbs and Hu Tri-NAb, the concentrations of Hu-αPDL1, Hu-α4-1BB, and Hu-αTIGIT were 10 μg/mL. After coincubation for 24 h, the cells were stained with propidium iodide, and the proportion of CFSE$^+$ PI$^+$ cells was detected using a BD FACSCelest™ flow cytometer (BD Biosciences). The data were analyzed using FlowJo software (version 10.0.7, TreeStar Inc., Ashland, USA).

### Evaluation of therapeutic efficacy in a murine xenograft model

Human peripheral blood mononuclear cells (PBMCs), mainly T and NK cells, were first injected intravenously into female NCG-hIL15 mice to rebuild the human immune system before constructing a subcutaneous HT-29 human colon cancer xenograft model. One week later, $2.0 \times 10^6$ HT-29 human colon cancer cells were inoculated subcutaneously into the right flanks of NCG-hIL15 mice in 75 μL of PBS, and the mice were treated intravenously with 1) the control, 2) Hu Tri-mAbs or 3) Hu Tri-NAb on Day 6 (doses of 2.5 mg/kg of Hu-αPDL1, Hu-α4-1BB, or Hu-αTIGIT), Day 9 (doses of 1.5 mg/kg of Hu-αPDL1, Hu-α4-1BB, or Hu-αTIGIT), and Day 12 (doses of 1.5 mg/kg of Hu-αPDL1, Hu-α4-1BB, or Hu-αTIGIT). The tumor volumes and body weights of the mice were monitored every two days. At the end of treatment, tumors were

collected for human cytokine detection using a QuickCyto® Human IFN-γ ELISA kit (NeoBioscience), QuantiCyto® Human Perforin ELISA kit (NeoBioscience), and QuantiCyto® Human Granzyme B ELISA kit (NeoBioscience) according to the manufacturer's protocol. In addition, the tumor tissues were sectioned and stained with rabbit anti-human CD3 (Servicebio) at 1:50 dilution and rabbit anti-human CD56 antibodies (Chinaway Biosciences) at 1:50 dilution for confocal laser scanning microscopy (CLSM) observation.

### Statistics and reproducibility

All experiments have been reproduced at least two times, and all attempts at replication were successful with self-consistent results. All results are presented as means ± standard deviations (s.d.), and differences for which $P < 0.05$ were considered significant. Significance levels were defined as n.s. (not significant, $P > 0.05$), $*P < 0.05$, $**P < 0.01$, $***P < 0.001$, and $****P < 0.0001$. $P$ values were analyzed using the Prism software package (GraphPad Prism 8.0). Student's $t$ test was used for two-group comparisons. One-way or two-way analysis of variance (ANOVA) and Tukey's post hoc test were used when more than two groups were compared (multiple comparisons). The survival benefit was analyzed using a log-rank (Mantel-Cox) test.

### Reporting summary

Further information on research design is available in the Nature Portfolio Reporting Summary linked to this article.

## Data availability

All data are available within the Article, Supplementary Information, or available from the authors upon request. RNA sequencing data have been deposited in NCBI Sequence Read Archive (SRA) under the accession code: SRR26390479, SRR26390478, SRR26390477, SRR26390476, SRR26390475, SRR26390474, SRR26390473, SRR26390472, SRR26390471, SRR26390470, SRR26390469, SRR26390468, SRR26390467, SRR26390466, SRR26390465. Source data are provided in this paper. Source data are provided with this paper.

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

## Acknowledgements
This work was supported by the National Key R&D Program of China (2021YFB3800900, S.S; 2022YFC3401400, J.W.), the National Natural Science Foundation of China (52130301, J.W.; 32071380, S.S.; 32271444, S.S.), the Science and Technology Program of Guangzhou, China (202103030004, J.W.), and the Guangdong Basic and Applied Basic Research Foundation (2022B1515020058, S.S.).

## Author contributions
S.S. and J.W. conceptualized, conceived and designed experiments. Q.-N.Y., L.Z., J.L., D.-K.Z., T.-Y.T., S.-Y.Y., H.L. X.-Y.H. and Z.-T.C. performed experiments. Q.-N.Y., L.Z., Y.-N.F. and S.S. analyzed data. Q.-N.Y., S.S. and J.W. wrote the manuscript. All authors discussed and edited the manuscript content.

## Competing interests
The authors declare no competing interests.
