## [Peer Review File · Nature Communications]

Orchestrating NK and T cells via tri-specific Nano-Antibodies for synergistic antitumor immunityREVIEWER COMMENTS

Reviewer #1 (Remarks to the Author):

Ye and collaborators show in this manuscript the therapeutic efficacy of Tri-specific nano-antibodies to improve CD8/NK cell-mediated anti-tumor responses. The use of novel albumin/polyester nanoparticles to improve biocompatibility, NP half-life and stability is interesting. In addition, the fact that this system allows the delivery of multiple mAbs to trigger both tumor cell death and the modulation of effector cells provides a very versatile strategy with potential clinical translation. Overall, the study is well done and the results obtained support the findings. However, some questions in terms of dosage, experimental replication and mechanism of action need to be addressed.

1. It is not clear what is the actual dosage of Tri-NABs that is being used in each experimental setting and if these dosages are equivalent to the dosage used for controls of free mAbs. In the material and methods section, the authors mention that the APCNs are mixed with the three mAbs at 1:1:1 (w/w), but what is the actual dosage of Tri-Nabs used after incubation? For in vitro experiments, free 5-10ug/mL of each mAb is used as control, but is this dose equivalent to the amount of mAb present in the Tri-NABs. In figure 3A it looks that the amount used of free mAb and tri-Nab binds to a similar degree to recombinant PD-L1, CD137 and NKG2A, however, the actual amount of Tri-NAb and free mAbs are not mentioned. In the material and methods' section (page 25 line 3), it states that a series of concentrations of free mAb (α PDL1, α 4-1BB, 4 or α NKG2A) or Tri-NAb were added to 5ug/mL recombinant proteins coated plates, however the display on Figure 3A is optical density (for binding efficiency) against recombinant proteins concentration (mM) and not antibody/NPAb concentrations. Is that correct? In addition, in figure 6 the authors checked different Tri-Nab doses (3.75, 6.25 and 7.5mg/mL of different concentrations for anti-PDL1/41BB/NKG2A used to generate the Tri-NABs). They said that in Figure 6i they halved the dose of anti-CD137 to 6.5mg/kg (page 15 line 5), however, according with the data showed in figure 6, the maximal dose used for anti-CD137 to generate tri-NABs in this figure was 2.5 mg/kg. Was higher dose used in previous experiments, because the dose used for free mAbs was 2.5 mg/kg each? Please clarify it.

2. Are the three mAb binded to the Tri-Nab at equal amount? How do the authors know that the Tri-NABs binds to the T/NK cells with anti-NKG2A and anti-CD137 present in one Tri-Nab as displayed in Figure 1 (and supplementary Figure 21) and not that NKG2A blockade and CD137 stimulation is accomplished by different Tri-NABs? Have the authors checked the immunological synapse of NK and/or T cells after tri-NABs to determine if both molecules are attached to their corresponding mAbs simultaneously from one or more nanoparticles? Determine the level of SHP1, ERK/NFkB phosphorylation of Tri-NABs-treated NK/T cells after PD-L1+Qa1b+ tumor cells exposure in comparison with single and bi-NABs-treated NK/T cells would also help to see if both pathways are activated at higher degree if the three mAb are present when compared to control, single and double NPs.

3. PD-L1 is also highly upregulated on both T and NK cells (PMID: 32117218, PMID: 35417187, PMID: 32152508). Did they author evaluated if the PD-L1 NP could bind to the effector cells? If so, did the binding was similar/better/worse than the binding to PDL1 expressed on tumor cells? Could this binding lead to effector cell fratricide?

4. In Figure 3g-h, it is quite surprising that no NK cells or T cells are seen in the isotype or free tri-mAbs cultures after having added 2.5×10^5 NK and CD8 T cells because binding through other NK/T cell receptors could be possible (like CEACAM, Fas, TRAIL, NKG2D, etc) as MC38 and B16 can express ligands for these pathways specially after IFN γ treatment (PMID: 22143889, PMID: 35840162). It is also surprising to see barely no NK cells and T cells in the NP anti-PDL1/41BB and NK anti-PDL1/NKG2A controls respectively since 41BB and NKG2A are expressed in both cell types as well. Please explain.

5. Depletion studies to determine the role of CD8 T cells and NK cells should be done to evaluate the importance of each immune cell type efficacy in the anti-tumor response mediated by Tri-NABs using single (anti-CD8b or anti-NK1.1) and double depletion.

6. In addition, a recent study demonstrated that the intratumoral administration of NK cells along with anti-NKG2A blockade mediated the anti-tumor response through cDC1 and CD8 T cells, but not endogenous NK cells (PMID: 37782273). One would expect that CD8 T cells dominate the anti-tumor response mediated by Tri-NABs despite triggering NK cells as well, given the low prevalence of NK cells within the TME of MC38 cells. Studies using RAG1 deficient mice (lack of B and T cells) could help to unveil the anti-tumor response of Tri-Nab dependent on NK cells. If better anti-tumor response by NK is proven, would the authors expect NK cell-mediated immunogenic cell death to increase CD8 T cell anti-tumor response in your experimental settings?

7. How do the authors reconcile the data shown in supplementary figure 20 and 22? In supplementary figure 22, authors show how NK and CD8 T cells reduce the percentage of tumor cells alive upon treatment (higher cytotoxicity), whereas in supplementary figure 20, it looks like the mAb treatment only halts tumor growth as the percentage of survival cells stays at 120 over time.

8. Can the authors show a FMO or biological control for anti-NK1.1 in Supplementary figure 25 or Figure 6f? Also, according to material and methods, a CD3 mAb was used in the mAb cocktail, thus it would be more appropriate gate first in CD3- cells to identify NK cells within the TME given that NKT cells do express NK1.1 as well. In addition, to identify CD8 and CD4 T cells, a prior gating on CD3 T cells is recommended.

9. Could the authors display the data shown in figure 6e and f as total number of CD8 T cells or NK cells per gram of tumor tissue? Given the marked reduction of tumor growth shown in figure 6b and c, provide the data this way would be a better way to determine if more effector cells are present in the TME upon treatment

10. In the discussion, a more detailed comparison of the data obtained in this manuscript with previous publication that target PD-L1, NKG2A, 41BB and /or TIGIT should be mention, given special attention to those strategies that combine stimulation and blockade of suppression and/or those strategies that simultaneously trigger three molecules like PMI 36071464 for example. Emphasize in why this approach is better than the others.

11. The authors need to mention in the figure legends and/or in the material and methods section how many times each experiment was done and if each figure panel it represents one, or it is representative of more independent experiments. This information is important to strengthen the reproducibility of the data.

Minor comments

1. The order and number of the Figures and supplementary figures should be mentioned in the text sequentially. Supplementary figure 17 is mentioned after Supplementary figure 13 (page 8 line 21), thus, it should be rewritten as Supplementary figure 14 instead of 17, and the rest of the figures named according to this change. Same for supplementary figure 21 and 22.

2. An introductory sentence should be added in page 13 line 20 to explain how the transcriptomic analysis was done.

3. Supplementary Figure 30 should not be detailed in the conclusion section. I would suggest moving this data that shows a safety profile for tri-Nab prior to describe the data of figure 7.

Reviewer #2 (Remarks to the Author):

The authors have developed nanoparticles coated with 3 antibodies targeting NKG2A, 41BB and PDL1, that they have called nano-antibodies or Tri-NABs. They have shown that these TriNABs exert anti-tumor effects both in vitro and in vivo in mouse tumor models. They have also shown that Tri-NABs coated with antibodies targeting human NKG2A, 41BB and PDL1 exert anti-tumoral effects in human tumor organoids and in humanized mouse models.

Although this represents an interesting technological development with a promising therapeutic potential, the study lacks a number of specificity controls and mechanistic analyses to warrant publication of the manuscript in its current form:

* The assays via flow cytometry, confocal imaging and ELISA in Figure 3 lack specificity controls. For example, do NP with anti-NKG2A and anti-41BB without anti-PDL1 still bind to the tumor cells? Can an excess of free antibodies block the binding of the nanoparticles to the cells and/or block the observed effects of the Tri-NAbs on NK/CD8 proliferation and binding to tumor cells? Also in the colon cancer organoids in Figure 7, such specificity controls are required.

* For the in vivo experiments in mice using the anti-mouse and the anti-human tri-NAbs, it is important to know how much of the effect is due to the tumor targeting properties of the Tri-NAbs as opposed to inherent properties of the NPs, by including a control group with NPs that have only anti- anti-NKG2A and anti-41BB without anti-PDL1.

* The authors assume that the superior inhibitory effect of Tri-NAb on mouse and human tumors may be attributed to the intensified adaptive and innate immune responses involving enhanced cytolytic activity of T and NK cells based on gene expressions and cytokine secretion data. However, to go beyond an observational study and prove that there is a causal relationship, the authors should perform depletion studies of NK cells and/or CD8 T cells to demonstrate which effector cell populations (NK cells, CD8 T cells or both) are required for the anti-tumor effect of the Tri-NAbs.

* Please indicate in Figure 4c what the orange and grey bars represent.

This study describes a new nanotechnology to conjugate anti-Fc antibody to the nanocore APCN and then add a combination of immunomodulatory antibodies, anti-PDL1, anti-4-1BB, and anti-NKG2A (or TIGIT) to form a therapeutic complex. Authors presented extensive biophysical characterization of the complex, its in vitro activity in activating NK and CD8 T cells, and bringing together NK and CD8 T cells to interact with tumor cells. Authors also presented impressive in vivo therapeutic data. In vivo safety profile was studied through serum biochemical characterization and organ histology and deemed to be safe. The study presented a novel technology to combine multiple antibodies for therapy. Conceptually is novel. Extensive in vitro and in vivo data was demonstrated for efficacy, mechanisms of action, and safety. However, there are several concerns that need to be addressed before can be accepted for publication.

Critique:

1. The title can be better worded for “Nano-Antibody” not to be confused with “nanobody”
2. Does the complex induce CRS in human PBMC? In vitro stimulation alone with TCR activation and in vivo data of serum cytokine would be informative. This is a safety concern with 4-1BB being a co-stimulatory receptor on T cells and NK cells.
3. What is ratio of the different antibody in the Tri-Nab being controlled to a consistency in each experiment? Is there a method to test the occupancy of each of the three antibodies? This has to be taken into consideration for the translational potential if the technology will be taken into use in the clinic.
4. Is the APCN biodegradable? How stable is Tri-Nab? What is the molecular weight of the Tri-Nab? Are there any histological evidence that the Tri-Nab penetrate into tumor tissues?
5. It is good that the authors presented PD-L1 expression in IFN γ stimulated tumor cells in vitro, which is a given. Is there any data to demonstrate PD-L1 expression in tumors in vivo by IHC or IF to support the MOA of the Tri-Nab in vivo?
6. Supplement Figure 25 – NK1.1 separates NK very well as a clear population. Current gating does not represent typical NK cell population and not convincing that the population is NK cell. Authors need to present a more convincing gating data, such as CD3. Vs. NK1.1 to define NK population.
7. Why in vitro data beautifully demonstrated the Tri-Nab brings tumor cells to simultaneously interact with NK and CD8 T cells, it would be more powerful to demonstrate the same interaction in tumors with multiplex IHC/IF.
8. Supplement Fig 30b, what are the authors intending to demonstrate?

Responses to the reviewers' concerns

We greatly appreciate the valuable comments of the three reviewers, helping us improve the quality of the manuscript. For clarity, the reviewers' comments are in black while our point-to-point answers are marked in blue. Additionally, the changes were highlighted (blue) in the revised manuscript and Supplementary Information for the convenience of reviewers.

Reviewer #1 (Remarks to the Author)

General Comments: Ye and collaborators show in this manuscript the therapeutic efficacy of Tri-specific nano-antibodies to improve CD8/NK cell-mediated anti-tumor responses. The use of novel albumin/polyester nanoparticles to improve biocompatibility, NP half-life and stability is interesting. In addition, the fact that this system allows the delivery of multiple mAbs to trigger both tumor cell death and the modulation of effector cells provides a very versatile strategy with potential clinical translation. Overall, the study is well done and the results obtained support the findings. However, some questions in terms of dosage, experimental replication and mechanism of action need to be addressed.

Response: We express our gratitude to the reviewer for recognizing our work and sincerely appreciate his/her insightful suggestions. We have elaborated on the concerns as shown below and hope the modifications to the manuscript will meet his/her criteria.

Comment 1: It is not clear what is the actual dosage of Tri-NAbs that is being used in each experimental setting and if these dosages are equivalent to the dosage used for controls of free mAbs. In the material and methods section, the authors mention that the APCNs are mixed with the three mAbs at 1:1:1 (w/w), but what is the actual dosage of Tri-Nabs used after incubation? For in vitro experiments, free 5-10 µg/mL of each mAb is used as control, but is this dose equivalent to the amount of mAb present in the Tri-NAbs? In figure 3A it looks that the amount used of free mAb and tri-Nab binds to a similar degree to recombinant PD-L1, CD137 and NKG2A,

however, the actual amount of Tri-NAb and free mAbs are not mentioned. In the material and methods' section (page 25 line 3), it states that a series of concentrations of free mAb (α PDL1, α 4-1BB, 4 or α NKG2A) or Tri-NAb were added to 5 μ g/mL recombinant proteins coated plates, however, the display on Figure 3A is optical density (for binding efficiency) against recombinant proteins concentration (mM) and not antibody/NAb concentrations. Is that correct?

Response: We sincerely apologize for the oversight in accurately describing the actual dosage of Tri-NAb. The equal probabilities of three mAbs binding to APCN@NA were confirmed at the outset of our study (data was not provided in the original manuscript). Tri-NAb was prepared by mixing APCN@NA with AF647-labeled α PDL1, AF488-labeled α 4-1BB, and AF555-labeled α NKG2A at a weight ratio of 1:1:1 at 4°C for 24 hours. The resulting Tri-NAb was separated through high-speed centrifugation, while the exact quantity of unbound mAbs remaining in the supernatant was quantified using high-performance liquid chromatography (HPLC), thereby enabling us to calculate the binding efficiency of each individual mAb to APCN@NA. As illustrated in Fig. R1 and Supplementary Fig. 16a, when the weight ratio between APCN@NA and total mAbs was 1:1, the binding efficiencies of α PDL1, α 4-1BB, and α NKG2A by APCN@NA were 89.5%, 91.6% and 91.8%, respectively. Based on these findings, it can be inferred that the final ratio of three mAbs on Tri-NAb was approximately 1:1:1. In subsequent experiments involving Tri-NAb preparation, unbound mAbs were removed and the term "dosage of Tri-NAb" refers to the combined quantity of all three mAbs present on Tri-Ab with an equal mass ratio unless otherwise specified. In the revised manuscript, we have revised relevant descriptions regarding the dosage of Tri-NAb.

Fig. R1. Binding efficiencies of α PDL1, α 4-1BB, and α NKG2A by APCN@NA. The weight ratio between APCN@NA and total mAbs was 1:1. Data are presented as means \pm s.d. (n = 3).

Additionally, we thank the reviewer for pointing out the error in Fig. 3a. The x-axis should represent a range of concentrations pertaining to free mAbs (α PDL1, α 4-1BB, or α NKG2A) or Tri-NAb rather than recombinant protein concentration (mM). We have meticulously corrected this in the revised manuscript.

Comment 2: In addition, in figure 6 the authors checked different Tri-Nab doses (3.75, 6.25 and 7.5mg/mL of different concentrations for anti-PDL1/41BB/NKG2A used to generate the Tri-NAbs). They said that in Figure 6i they halved the dose of anti-CD137 to 6.5 mg/kg (page 15 line 5), however, according with the data showed in figure 6, the maximal dose used for anti-CD137 to generate tri-NAbs in this figure was 2.5 mg/kg. Was higher dose used in previous experiments, because the dose used for free mAbs was 2.5 mg/kg each? Please clarify it.

Response: In all other animal studies, we administered a dosage of 2.5 mg/kg for each mAb, resulting in a total dose of 7.5 mg/kg. However, in the experiment depicted in Fig. 6i, mice were treated with a total Tri-NAb dosage of 3.75 mg/kg, equivalent to a dosage of 1.25 mg/kg for each mAb; or alternatively, a total Tri-NAb dosage of 6.25 mg/kg comprising of 2.5 mg/kg α PDL1, 1.25 mg/kg α 4-1BB, and another 2.5 mg/kg α NKG2A; or finally, a total Tri-NAb dosage of 7.5 mg/kg consisting of equal dosages (2.5mg/kg) for each mAb. We have included a detailed description in the revised Fig. 6i and in the Methods section of the revised manuscript.

Comment 3: Are the three mAb binded to the Tri-Nab at equal amount? How do the authors know that the Tri-NAbs binds to the T/NK cells with anti-NKG2A and anti-CD137 present in one Tri-NAb as displayed in Figure 1 (and supplementary Figure 21) and not that NKG2A blockade and CD137 stimulation is accomplished by different Tri-NAbs? Have the authors checked the immunological synapse of NK and/or T cells after tri-NAbs to determine if both molecules are attached to their corresponding mAbs simultaneously from one or more nanoparticles? Determine the level of SHP1, ERK/NFkB phosphorylation of Tri-NAbs-treated NK/T cells after PD-L1+Qa1b+ tumor cells exposure in comparison with single and bi-NAbs-treated NK/T cells would also help to see if both pathways are activated at higher degree if the three mAb are present when compared to control, single and double NPs.

Response: Thanks for the insightful comment. In the original manuscript, we demonstrated that APCN@NA exhibited comparable binding affinities towards Rat IgG2a or Rat IgG2b mAbs (Fig. 2m-n). Furthermore, we have shown α PDL1 (Rat IgG2b subtype), α 4-1BB (Rat IgG2a subtype), and α NKG2A (Rat IgG2a subtype) bind to APCN@NA with high efficiencies of 89.5%, 91.6%, and 91.8%, respectively, indicating a nearly equal affinity of APCN@NA towards these three mAbs (Fig. R1 and Supplementary Fig. 16a). Based on these findings, it can be inferred that the distribution of the three mAbs on an APCN@NA is random. Therefore, in the schematic diagram, we depict an equal number of all three mAbs attached to one APCN@NA.

We express our gratitude to the reviewer for highlighting the inadequacy of our schematic depiction. Undoubtedly, the augmented interaction between tumor cells and NK/T cells mediated, as well as the enhanced activation of NK/T cells, is not achieved by a single Tri-NAb alone but rather through the synergistic action of multiple Tri-NAb. Considering the stochastic distribution of 4-1BB and NKG2A on NK/T cells, it is plausible that α 4-1BB and α NKG2A from one Tri-NAb may simultaneously recognize and engage their respective targets; however, it is also feasible that only α 4-1BB or α NKG2A from one Tri-NAb interacts with its target. We observed that Tri-NAb mediated the formation of immune synapses between NK/T

cells and tumor cells (Fig. R2), however, discerning whether synapse formation is attributed to simultaneous attachment of 4-1BB and NKG2A to their corresponding mAbs from one or more Tri-NAb remains challenging; we suspect that both possibilities exist. To avoid any potential misunderstanding, we have revised related schematic diagrams and added more descriptions in the revised manuscript.

Fig. R2. Representative CLSM images showing the formation of immune synapses between NK/T cells and tumor cells. NK or CD8⁺ T cells were pretreated with Tri-NAb for activation and then incubated with the tumor cells, followed by fixation and staining with AF488-phalloidines.

Additionally, per the reviewer's suggestion, western blot analysis was performed to assess the activation of NK and CD8⁺ T cells by examining the levels of SHP-1 and ERK/NFκB phosphorylation. Stimulated NK or CD8⁺ T cells were co-incubated with PD-L1⁺Qa-1b⁺ MC38 tumor cells and subsequently treated with IgG control, Tri-mAbs, NP_{αPDL1+α4-1BB}, NP_{αPDL1+αNKG2A}, NP_{α4-1BB+αNKG2A}, or Tri-NAb. After 24 hours of co-incubation, NK or CD8⁺ T cells were collected for western blot analysis. As depicted in Fig. R3, following treatment with Tri-NAb, both NK and CD8⁺ T cells exhibited enhanced NFκB phosphorylation levels while demonstrating a significant decrease in SHP-1 levels compared to other groups. These findings indicate that both pathways were activated to a greater extent in the Tri-NAb-treated group. Related data

was also included in **Supplementary Fig. 25** in the revised **Supplementary Information**.

Fig. R3. Levels of SHP-1, ERK/NFκB phosphorylation of NK and CD8⁺ T cells after PD-L1⁺Qa-1^{b+} tumor cells exposure. The stimulated NK or CD8⁺ T cells were co-incubated with IFN-γ-stimulated MC38 tumor cells, and then were treated with IgG control, Tri-mAbs, NP_{αPDL1+α4-1BB}, NP_{αPDL1+αNKG2A}, NP_{α4-1BB+αNKG2A}, or Tri-NAb, where the concentration of αPDL1, α4-1BB, or αNKG2A was 10 μg/mL respectively, and the concentration of mAbs was 30 μg/mL. After co-incubation for 24 h, NK or CD8⁺ T cells were collected and subjected to western blot using Jess Automated Western Blot System (ProteinSimple, Bio-Techne).

Comment 4: PD-L1 is also highly upregulated on both T and NK cells (PMID: 32117218, PMID: 35417187, PMID: 32152508). Did they author evaluated if the PD-L1 NP could bind to the effector cells? If so, did the binding was similar/better/worse than the binding to PDL1 expressed on tumor cells? Could this binding lead to effector cell fratricide?

Response: Thank you for your insightful comment. Flow cytometry analysis revealed a minimal binding of NP_{αPD-L1} to NK or CD8⁺ T cells, which was significantly overshadowed by its remarkable affinity for tumor cells (Fig. R4). Per the reviewer's suggestion, we investigated to determine whether Tri-NAb induces fratricide of NK/T cells themselves. To accomplish this, we incubated NK or/and CD8⁺ T with Tri-NAb and propidium iodide (for labeling apoptotic cells), and observed cell apoptosis utilizing an advanced high-content assay system. As illustrated in **Supplementary Video 1**, no evidence of fratricidal behavior among NK cells was observed following

treatment with Tri-NAb. Similarly, CD8⁺ cells did not exhibit any such tendencies as shown in Supplementary Video 2. Furthermore, even when NK and CD8⁺ T cells were co-incubated together, there was still no indication of fratricide occurring between these two cell populations as demonstrated in Supplementary Video 3. The negligible occurrence of effector cell fratricide may be attributed to the relatively weak cellular interaction facilitated by Tri-NAb and their inherent self-protection mechanism. We have included related description and experimental details in the revised manuscript.

Fig. R4. The binding of Cy5-labeled NP_{αPDL1} to MC38 cells, NK cells, and CD8⁺ T cells.

Comment 5: In Figure 3g-h, it is quite surprising that no NK cells or T cells are seen in the isotype or free tri-mAbs cultures after having added 2.5×10^5 NK and CD8 T cells because binding through other NK/T cell receptors could be possible (like CEACAM, Fas, TRAIL, NKG2D, etc) as MC38 and B16 can express ligands for these pathways specially after IFN γ treatment (PMID: 22143889, PMID: 35840162). It is also surprising to see barely no NK cells and T cells in the NP anti-PDL1/41BB and NK anti-PDL1/NKG2A controls, respectively, since 41BB and NKG2A are expressed in both cell types as well. Please explain.

Response: Prior to imaging, we utilized pre-cooled 1×PBS to wash the cells several times to diminish the weak contact between NK/T cells and tumor cells facilitated by other ligands. However, nanoparticles-treated NK/T cells exhibited enhanced interactions with tumor cells growing on the petri dishes, rendering them less susceptible to being washed away. Through extensive observation of various fields and quantification of co-localized regions between NK/T cells and tumor cells, it was observed that Tri-NAbs-treated group displayed a significantly augmented abundance of tightly adhered NK/T cells to tumor cells (as illustrated in Fig. 3i).

Comment 6: Depletion studies to determine the role of CD8 T cells and NK cells should be done to evaluate the importance of each immune cell type efficacy in the anti-tumor response mediated by Tri-NAbs using single (anti-CD8b or anti-NK1.1) and double depletion.

Response: Thanks to the reviewer for the insightful comments. Per the reviewer's suggestion, we conducted experiments to evaluate the involvement of CD8⁺ T and NK cells in Tri-NAb-mediated suppression of tumor growth by selectively depleting these cell populations using specific antibodies (Fig. R5a). As depicted in Fig. R5b-d, depletion of CD8⁺ T cells significantly impeded the remarkable antitumor efficacy of Tri-NAb, as evidenced by the tumor growth curves; similarly, depletion of NK cells had a substantial impact on the antitumor effect of Tri-NAb, particularly on the complete remission rate (CR). As anticipated, simultaneous depletion of both CD8⁺ T and NK cells mediated by anti-CD8 and anti-NK1.1 antibodies resulted in a complete loss of the antitumor potency exhibited by Tri-NAb. These findings underscored the crucial roles played by both CD8⁺ T and NK cells in mediating the anticancer response elicited by Tri-NAbs, with CD8⁺ T cells being predominant. Related data was also included in Supplementary Fig. 32 in the revised Supplementary Information.

Fig. R5. Both CD8⁺ T and NK cells played crucial roles in Tri-NAb-mediated tumor growth suppression. (a) Representative scatter plots of flow cytometry showing the proportion of CD8⁺ T/NK cells in the peripheral blood of mice after antibody-mediated depletion. (b) A subcutaneous MC38 tumor model was established in male C57BL/6 mice. According to the experimental protocol, CD8⁺ T and NK cells were depleted via intraperitoneal injection of 125 μ g anti-CD8 mAb or/and 150 μ g anti-NK1.1 mAb on specific days. Tri-NAb was administrated every three days for three repeats since the 8th day post-tumor inoculation; the injection dose of α PDL1, α 4-1BB, and α NKG2A was 2.5 mg/kg. Average (c) and individual (d) tumor growth curves of MC38 tumors after being treated with Tri-NAb with or without T cells or/and NK cells depletion. The statistical data are presented as the mean \pm s.d. (n = 5). Statistical significance was calculated by one-way ANOVA with Tukey's post hoc test. $****P < 0.0001$.

Comment 7: In addition, a recent study demonstrated that the intratumoral administration of NK cells along with anti-NKG2A blockade mediated the anti-tumor response through cDC1 and CD8 T cells, but not endogenous NK cells (PMID: 37782273). One would expect that CD8 T cells dominate the anti-tumor response mediated by Tri-NAbs despite triggering NK cells as well, given the low prevalence of NK cells within the TME of MC38 cells. Studies using RAG1 deficient mice (lack of B and T cells) could help to unveil the anti-tumor response of Tri-Nab dependent on NK cells. If better anti-tumor response by NK is proven, would the authors expect NK cell-mediated immunogenic cell death to increase CD8 T cell anti-tumor response in your experimental settings?

Response: Per the reviewer’s suggestion, we conducted experiments to elucidate the role of NK cells in Tri-NAb-mediated anti-tumor response and utilized Rag1-deficient (Rag1^{-/-}) mice as our experimental model. Initially, flow cytometry analysis revealed a remarkable absence of T cells alongside a significant augmentation in the proportion of NK cells in the peripheral blood of Rag1-deficient mice (Fig. R6a). Subsequently, we established a subcutaneous MC38 tumor model and administered Tri-NAb treatment according to our experimental design (Fig. R6b). Although there was mild control over colon cancer growth in Rag1-deficient mice following Tri-NAb treatment (Fig. R6c), it was not as profoundly suppressed as observed in immunocompetent C57BL/6 mice (Fig. 4f). These findings suggest that the antitumor response elicited by Tri-NAb is moderately reliant on NK cells, with T cells play a more crucial role. The revised manuscript has provided the corresponding data as Supplementary Fig. 33.

Fig. R6. Tri-NAb mildly controlled tumor progression in Rag1-deficient mice. (a) Representative flow cytometry scatter plots show the proportion of CD8⁺ T and NK cells in the peripheral blood of immunocompetent C57BL/6 mice and Rag1-deficient mice. (b) Experimental scheme of subcutaneous MC38 tumor model in female Rag1-deficient mice. Tri-NAb with equivalent doses of α PDL1, α 4-1BB, and α NKG2A (2.5 mg/kg each) was intravenously (i.v.) administered via the tail on days 8, 11, and 14 post MC38 tumor inoculation. (c) Tumor growth curves of MC38 tumors. The statistical data are presented as the mean \pm s.d. (n = 5). Statistical significance was calculated via paired t-test with two-tailed. *** $P < 0.001$.

Comment 8: How do the authors reconcile the data shown in supplementary figure 20 and 22? In supplementary figure 22, authors show how NK and CD8 T cells reduce the percentage of tumor cells alive upon treatment (higher cytotoxicity), whereas in supplementary figure 20, it looks like the mAb treatment only halts tumor growth as the percentage of survival cells stays at 120 over time.

Response: We express our gratitude for the meticulous review. Following a comprehensive and repeated examination of the data, we have identified that the disparity in outcomes arose due to the inadvertent inclusion of all co-cultured cells (including NK/T cells) in original Supplementary Fig. 20 when quantifying tumor cell viability. Conversely, the original Supplementary Fig. 22 accurately assessed tumor cell viability by measuring the exclusive bioluminescence signals emitted by luciferase within MC38 tumor cells. We sincerely apologize for employing an inappropriate analysis method in original Supplementary Fig. 20 and reanalyzed the high-content data to determine tumor cell viability more precisely by integrating both tumor cell phenotypes and fluorescence intensity labeling on tumor cells, thereby providing a more precise reflection of the actual extent of tumor cell eradication within 24 hours (as illustrated in Fig. R7 and revised Supplementary Fig. 24). Similarly, Tri-NAb exhibited approximately a 50% efficacy in inducing apoptosis among tumor cells after a duration of 24 hours, while the viability of tumor cells in Tri-mAb group decreased slightly with the extension of culture time.

Fig. R7. High-content analysis platform for monitoring the viability of tumor cells. PKH26 labelled-MC38 tumor cells (5.0×10^3) were seeded in CellCarrier-96 Ultra Microplates (PerkinElmer) and allowed to attach overnight. The stimulated CD8⁺ T cells (5.0×10^4) and

NK cells (5.0×10^4) were then added. Co-cultured cells were treated with IgG Control, Tri-mAbs, NP $_{\alpha\text{PDL1}+\alpha\text{NKG2A}}$, NP $_{\alpha\text{PDL1}+\alpha\text{4-1BB}}$ or Tri-NAb; the concentration of αPDL1 , $\alpha\text{4-1BB}$, or αNKG2A was 10 $\mu\text{g}/\text{mL}$ respectively, and the concentration of IgG was 30 $\mu\text{g}/\text{mL}$. The plates were incubated at 37 °C and 5% CO₂, and cell images were continuously acquired every 30 min using Operetta CLS™ High-Content Analysis System (PerkinElmer) for 24 h. The viability of PKH26 labelled-MC38 tumor cells was evaluated using Harmony® high-content analysis software based on cellular phenotypes and fluorescence distribution parameters.

Comment 9: Can the authors show a FMO or biological control for anti-NK1.1 in Supplementary figure 25 or Figure 6f? Also, according to material and methods, a CD3 mAb was used in the mAb cocktail, thus it would be more appropriate gate first in CD3- cells to identify NK cells within the TME given that NKT cells do express NK1.1 as well. In addition, to identify CD8 and CD4 T cells, a prior gating on CD3 T cells is recommended.

Response: To further explore the anti-tumor advantages of Tri-NAb and obtain a more precise analysis of tumor-infiltrating immune cells post-treatment, we incorporated the NP $_{\alpha\text{4-1BB}+\alpha\text{NKG2A}}$ group and repeated the *in vivo* antitumor experiment using the B16-F10 melanoma model. As depicted in Fig. R8, Tri-NAb exhibited significant inhibition of tumor growth compared to the other groups. Tumor tissues were collected at the end of treatment for flow cytometric analysis. The gating strategy recommended by the reviewer was employed to analyze tumor-infiltrating lymphocytes. As illustrated in Fig. R9, the NK cell population was identified as CD45⁺CD3⁻NK1.1⁺, and Fluorescence Minus One (FMO) control for anti-NK1.1 was also included in the flow cytometric analysis. Additionally, CD45⁺CD3⁺ cells were gated to analyze CD8⁺ and CD4⁺ T cells. The revised gating strategy was included as Supplementary Fig. 35 in the revised Supplementary Information.

Fig. R8. Tri-NAb effectively inhibited the progression of mouse melanoma. (a) Experimental scheme of subcutaneous B16-F10 (4.0×10^5) mouse melanoma model in male C57BL/6 mice. Different formulations with equivalent doses of α PDL1, α 4-1BB, and α NKG2A (2.5 mg/kg each) were intravenously (i.v.) administered via the tail on days 6, 9, and 12 after B16-F10 tumor inoculation. Individual (b) and average (c) tumor growth kinetics in different formulations. Growth curves represent means \pm s.d. ($n = 6$). Representative scatter plots of flow cytometry showing the number of NK cells as a percentage of CD3⁺ cell population (d) or CD8⁺ T cells as a percentage of CD3⁺ cell population (g) in the tumor after different treatments as indicated. Quantitative results of the number of NK (e) or CD8⁺ T cells (h) as a percentage of the total CD45⁺ cell population in the tumor. Quantitative results of the total number of NK (f) or CD8⁺ T cells (i) per gram of tumor tissue. n.s., not significant. FACS data are presented as means \pm s.d. ($n = 4$). Statistical significance was calculated by one-way ANOVA with Tukey's post hoc test. * $P < 0.05$; ** $P < 0.01$; *** $P < 0.001$; **** $P < 0.0001$.

Fig. R9. Gating strategy for tumor-infiltrating lymphocyte.

Comment 10: Could the authors display the data shown in figure 6e and f as total number of CD8 T cells or NK cells per gram of tumor tissue? Given the marked reduction of tumor growth shown in figure 6b and c, provide the data this way would be a better way to determine if more effector cells are present in the TME upon treatment.

Response: Thanks for the meaningful comment. As shown in Fig. R8f and 8i, Tri-NAb treatment significantly increased the total number of both NK and CD8⁺ T cells per gram of tumor tissue, surpassing the other groups, where either NK cells or CD8⁺ T cells were predominantly increased. This underscores the remarkable advantages of employing an orchestrated combination therapy with Tri-NAb. The data have been provided in Fig. 6 and Supplementary Fig. 34 in the revised manuscript and Supplementary Information.

Comment 11: In the discussion, a more detailed comparison of the data obtained in this manuscript with previous publication that target PD-L1, NKG2A, 41BB and /or TIGIT should be mentioned, given special attention to those strategies that combine stimulation and blockade of suppression and/or those strategies that simultaneously trigger three molecules like 36071464 for example. Emphasize in why this approach is better than the others.

Response: Thanks for the insightful comment. We have added a more detailed discussion in the revised manuscript.

Comment 12: The authors need to mention in the figure legends and/or in the material and methods section how many times each experiment was done and if each figure panel it represents one, or it is representative of more independent experiments. This information is important to strengthen the reproducibility of the data.

Response: We have added more detailed description of experimental in the figure legends and the Methods section in the revised manuscript.

Comment 13: The order and number of the Figures and supplementary figures should be mentioned in the text sequentially. Supplementary figure 17 is mentioned after Supplementary figure 13 (page 8 line 21), thus, it should be rewritten as Supplementary figure 14 instead of 17, and the rest of the figures named according to this change. Same for supplementary figure 21 and 22.

Response: We deeply apologize for our oversight and have meticulously reviewed and revised it.

Comment 14: An introductory sentence should be added in page 13 line 20 to explain how the transcriptomic analysis was done.

Response: Thanks for the comment. In the revised manuscript, we have carefully supplemented the implementation method of transcriptomic analysis.

Comment 15: Supplementary Figure 30 should not be detailed in the conclusion section. I would suggest moving this data that shows a safety profile for tri-Nab prior to describe the data of figure 7.

Response: Thanks to the reviewer for the friendly and meaningful suggestion. We have moved this data prior to Fig. 7 in the revised manuscript.

Reviewer #2 (Remarks to the Author)

General Comments: The authors have developed nanoparticles coated with 3 antibodies targeting NKG2A, 41BB and PDL1, that they have called nano-antibodies or Tri-NAbs. They have shown that these TriNAbs exert anti-tumor effects both in vitro and in vivo in mouse tumor models. They have also shown that Tri-NAbs coated with antibodies targeting human NKG2A, 41BB and PDL1 exert anti-tumoral effects in human tumor organoids and in humanized mouse models. Although this represents an interesting technological development with a promising therapeutic potential, the study lacks a number of specificity controls and mechanistic analyses to warrant publication of the manuscript in its current form.

Response: We express our gratitude to the reviewer for recognizing our work and sincerely appreciate his/her insightful suggestions. We have elaborated on the concerns as shown below and hope the modifications to the manuscript will meet his/her criteria.

Comment 1: The assays via flow cytometry, confocal imaging and ELISA in Figure 3 lack specificity controls. For example, do NP with anti-NKG2A and anti-41BB without anti-PDL1 still bind to the tumor cells? Can an excess of free antibodies block the binding of the nanoparticles to the cells and/or block the observed effects of the Tri-NAbs on NK/CD8 proliferation and binding to tumor cells? Also in the colon cancer organoids in Figure 7, such specificity controls are required.

Response: Thanks for the meaningful comment. Following the reviewer's suggestion, we incorporated the NP _{α 4-1BB+ α NKG2A} group (NP with α 4-1BB and α NKG2A and without α PDL1) into our experiments and repeated them accordingly. Utilizing flow cytometry analysis, we observed minimal binding of NP _{α 4-1BB+ α NKG2A} to tumor cells but found a highly efficient association with NK/T cells. (Fig. R10 and revised Supplementary Fig. 18, the third group). When tumor cells were pre-incubated with excess α PDL1, or NK/T cells with excess α 4-1BB and α NKG2A, there was a significant reduction in Tri-NAb binding to these cells, indicating a high likelihood of binding through targets with mAbs on Tri-NAb (Fig. R10 and revised Supplementary

Fig. 18, the fourth and fifth groups). As anticipated, treatment with NP_{α4-1BB+αNKG2A} also markedly promoted NK/T cell proliferation similar to other NPs immobilizing α4-1BB (Fig. R11 and revised Supplementary Fig. 19). However, through CLSM observation, it was evident that the ability of NP_{α4-1BB+αNKG2A} to enhance interactions between NK/T cells and tumor cells was notably weaker compared to Tri-NAb (Fig. R12b and revised Supplementary Fig. 20), presumably due to decreased recognition and binding capacity towards tumor cells caused by deficiency in αPDL1. These feeble cellular interactions may elucidate why levels of IFN-γ, granzyme B, and perforin released by NP_{α4-1BB+αNKG2A}-treated NK/T cells were not as robust as those released by the Tri-NAb-treated cells, alongside the moderate capability of NP_{α4-1BB+αNKG2A}-treated NK/T cells to eliminate tumor cells (Fig. R12c-f and revised Supplementary Fig. 23). Similarly, we detected the mild killing activity of NP_{α4-1BB+αNKG2A}-mediated NK/T cell on patient-derived colon cancer organoids (Fig. R13 and revised Fig. 7f-g). These data highlighted the importance of αPDL1-mediated tumor cell recognition and bridging for Tri-NAb to exert excellent anti-tumor efficacy.

Fig. R10. The binding of NPs to tumor cells, NK cells or CD8⁺ T cells and tumor cells. Representative FACS histograms and statistics of MC38 murine colon cancer cells (a), NK (b), and CD8⁺ T cells (c) after incubation with PBS, Cy5-labeled NP $_{\alpha PDL1}$, NP $_{\alpha 4-1BB + \alpha NKG2A}$, Tri-NAb, or Tri-NAb with mAbs pretreatment. Data were presented as mean \pm s.d. (n = 3). Statistical significance was calculated by one-way ANOVA with Tukey's post hoc test. **** $P < 0.0001$.

Fig. R11. Tri-NAb significantly promoted the proliferation of NK and CD8⁺ T cells. Representative FACS histograms of CFSE-labeled NK cells (a) and CFSE-labeled CD8⁺ T cells (b) after incubation with 1) control; 2) $\alpha 4$ -1BB; 3) NP $_{\alpha 4-1BB}$; 4) NP $_{\alpha PDL1+\alpha 4-1BB}$; 5) NP $_{\alpha 4-1BB+\alpha NKG2A}$; and 6) Tri-NAb. Data were presented as mean \pm s.d. (n = 3). Statistical significance was calculated by one-way ANOVA with Tukey's post hoc test. ** $P < 0.01$, **** $P < 0.0001$.

Fig. R12. Tri-NAb effectively activated both NK and CD8⁺ T cells *in vitro*. (a) Scheme of the interaction and cytotoxicity of NK and CD8⁺ T cells to MC38 tumor cells after incubation with 1) control; 2) $\alpha PDL1$ & $\alpha 4$ -1BB & $\alpha NKG2A$ (Tri-mAbs); 3) NP $_{\alpha PDL1+\alpha 4-1BB}$; 4) NP $_{\alpha PDL1+\alpha NKG2A}$; 5) NP $_{\alpha 4-1BB+\alpha NKG2A}$; 6) Tri-NAb. (b) Representative confocal images of CD8⁺

T and NK cells interacting with MC38 tumor cells after being treated with NP_{α4-1BB+αNKG2A}. (c) Representative confocal images of Tri-NAb-mediated NK and CD8⁺ T cells interacting with MC38 tumor cells after pre-blocking tumor cells with excess αPDL1, or NK/T cells with excess α4-1BB and αNKG2A. The release levels of IFN-γ (d), granzyme B (e), and perforin (f) in the supernatant of the co-incubation system were examined by ELISA. (g) Viability of MC38-luc cells in the co-incubation system. MC38-luc, MC38 cells expressing the luciferase gene. Data were presented as mean ± s.d. (n = 3). Statistical significance was calculated by one-way ANOVA with Tukey's post hoc test. *P < 0.05; **P < 0.01; ***P < 0.001; ****P < 0.0001.

Fig. R13. Hu Tri-NAb exhibited potent killing activity on patient-derived colon cancer organoids. Representative scatter plots of flow cytometry (a) and corresponding quantification results (b) showing the number of CFSE⁺ PI⁺ cells as a percentage of total cells. Data are presented as means ± s.d. (n = 4). Statistical significance was calculated by one-way ANOVA with Tukey's post hoc test. *P < 0.05; ****P < 0.0001.

Comment 2: For the in vivo experiments in mice using the anti-mouse and the anti-human tri-NAbs, it is important to know how much of the effect is due to the tumor targeting properties of the Tri-NAbs as opposed to inherent properties of the NPs, by including a control group with NPs that have only anti- anti-NKG2A and anti-41BB without anti-PDL1.

Response: Thanks for the insightful comment. Per the reviewer's suggestion, we incorporated the NP_{α4-1BB+αNKG2A} group and repeated the antitumor experiment using the B16-F10 melanoma model. Tumor volumes were monitored throughout treatment while tumor-infiltrating NK and CD8⁺ T cells were analyzed at the end of treatment. As illustrated in Fig. R8, Tri-NAb significantly impeded tumor growth and modulated the tumor microenvironment more effectively than other groups, including NP_{α4-1BB+αNKG2A}. Although NP_{α4-1BB+αNKG2A} theoretically exerts a similar impact on

NK/T cell activation and proliferation as Tri-NAb does, it is noteworthy that Tri-NAb exhibits a superior anti-tumor effect compared to NP_{α4-1BB+αNKG2A}. We hypothesize that this discrepancy arises from two distinct characteristics of Tri-NAb when contrasted with NP_{α4-1BB+αNKG2A}: firstly, by leveraging anti-PDL1 mAbs assistance, Tri-NAb can bind PDL1-positive tumor cells more efficiently, thereby enhancing their accumulation within tumors and prolonging their residence time; secondly, Tri-NAb substantially augments interactions between NK/T cells and tumor cells, consequently increasing granzyme B and perforin release by NK/T cells for enhanced killing of neighboring tumor cells. Revised Fig. 6 and Supplementary Fig. 34 containing relevant data have been included in the revised manuscript.

Fig. R8. Tri-NAb effectively inhibited the progression of mouse melanoma. (a) Experimental scheme of subcutaneous B16-F10 (4.0×10⁵) mouse melanoma model in male C57BL/6 mice. Different formulations with equivalent doses of αPDL1, α4-1BB, and αNKG2A (2.5 mg/kg each) were intravenously (i.v.) administered via the tail on days 6, 9, and 12 after B16-F10 tumor inoculation. Individual (b) and average (c) tumor growth kinetics in different formulations. Growth curves represent means ± s.d. (n = 6). Representative scatter plots of flow cytometry showing the number of NK cells as a percentage of CD3⁺ cell population (d) or CD8⁺ T cells as a percentage of CD3⁺ cell population (g) in the tumor after

different treatments as indicated. Quantitative results of the number of NK (e) or CD8⁺ T cells (h) as a percentage of the total CD45⁺ cell population in the tumor. Quantitative results of the total number of NK (f) or CD8⁺ T cells (i) per gram of tumor tissue. n.s., not significant. FACS data are presented as means ± s.d. (n = 4). Statistical significance was calculated by one-way ANOVA with Tukey's post hoc test. **P* < 0.05; ***P* < 0.01; ****P* < 0.001; *****P* < 0.0001.

Comment 3: The authors assume that the superior inhibitory effect of Tri-NAb on mouse and human tumors may be attributed to the intensified adaptive and innate immune responses involving enhanced cytolytic activity of T and NK cells based on gene expressions and cytokine secretion data. However, to go beyond an observational study and prove that there is a causal relationship, the authors should perform depletion studies of NK cells and/or CD8 T cells to demonstrate which effector cell populations (NK cells, CD8 T cells or both) are required for the anti-tumor effect of the Tri-NAbs.

Response: Per the reviewer's suggestion, we conducted experiments to evaluate the involvement of CD8⁺ T and NK cells in Tri-NAb-mediated suppression of tumor growth by selectively depleting these cell populations using specific antibodies (Fig. R5a). As depicted in Fig. R5b-d, depletion of CD8⁺ T cells significantly impeded the remarkable antitumor efficacy of Tri-NAb, as evidenced by the tumor growth curves; similarly, depletion of NK cells had a substantial impact on the antitumor effect of Tri-NAb, particularly on the complete remission rate (CR). As anticipated, simultaneous depletion of both CD8⁺ T and NK cells mediated by anti-CD8 and anti-NK1.1 antibodies resulted in a complete loss of the antitumor potency exhibited by Tri-NAb. These findings underscored the crucial roles played by both CD8⁺ T and NK cells in mediating the anticancer response elicited by Tri-NAbs, with CD8⁺ T cells being predominant. Related data was also included in Supplementary Fig. 32 in the revised Supplementary Information.

Fig. R5. Both CD8⁺ T and NK cells played crucial roles in Tri-NAb-mediated tumor growth suppression. (a) Representative scatter plots of flow cytometry showing the proportion of CD8⁺ T/NK cells in the peripheral blood of mice after antibody-mediated depletion. (b) A subcutaneous MC38 tumor model was established in male C57BL/6 mice. According to the experimental protocol, CD8⁺ T and NK cells were depleted via intraperitoneal injection of 125 μ g anti-CD8 mAb or/and 150 μ g anti-NK1.1 mAb on specific days. Tri-NAb was administrated every three days for three repeats since the 8th day post-tumor inoculation; the injection dose of α PDL1, α 4-1BB, and α NKG2A was 2.5 mg/kg. Average (c) and individual (d) tumor growth curves of MC38 tumors after being treated with Tri-NAb with or without T cells or/and NK cells depletion. The statistical data are presented as the mean \pm s.d. (n = 5). Statistical significance was calculated by one-way ANOVA with Tukey's post hoc test. $****P < 0.0001$.

Comment 4: Please indicate in Figure 4c what the orange and grey bars represent.

Response: Thanks for the kind reminder. Fig. 4c illustrates the fluorescence quantitative statistics in major organs and tumor tissues 36 hours after *i.v.* administration of Cy5-labeled Tri-mAbs (represented by grey bars) or Tri-NAb (represented by orange bars). We have modified Fig. 4c in the revised manuscript.

Reviewer #3 (Remarks to the Author)

General Comments: This study describes a new nanotechnology to conjugate anti-Fc antibody to the nanocore APCN and then add a combination of immunomodulatory antibodies, anti-PDL1, anti-4-1BB, and anti-NKG2A (or TIGIT) to form a therapeutic complex. Authors presented extensive biophysical characterization of the complex, its in vitro activity in activating NK and Cd8 T cells, and bringing together NK and CD8 T cells to interact with tumor cells. Authors also presented impressive in vivo therapeutic data. In vivo safety profile was studied through serum biochemical characterization and organ histology and deemed to be safe. The study presented a novel technology to combine multiple antibodies for therapy. Conceptually is novel. Extensive in vitro and in vivo data was demonstrated for efficacy, mechanisms of action, and safety. However, there are several concerns that need to be addressed before can be accepted for publication.

Response: We express our gratitude to the reviewer for recognizing our work and sincerely appreciate his/her insightful suggestions. We have elaborated on the concerns as shown below and hope the modifications to the manuscript will meet his/her criteria.

Comment 1: The title can be better worded for “Nano-Antibody” not to be confused with “nanobody”.

Response: Thanks to the reviewer for the thoughtful suggestion. We have revised it in the revised manuscript.

Comment 2: Does the complex induce CRS in human PBMC? In vitro stimulation alone with TCR activation and in vivo data of serum cytokine would be informative. This is a safety concern with 4-1BB being a co-stimulatory receptor on T cells and NK cells.

Response: The reviewer posed a profoundly significant inquiry. As the cross-linking of 4-1BB, either by binding to 4-1BBL or agonist antibody, delivers a costimulatory signal for the activation and proliferation of NK/T-cells, as well as the release of

various cytokines. Therefore, the biosafety of Tri-NAb emerged as a pivotal concern in our study. It is indeed gratifying that no serious adverse events were observed during Tri-NAb treatment; this was substantiated by negligible weight loss throughout the course of treatment (revised Supplementary Fig. 30 and Fig. 36) and corroborated by post-treatment serum biochemical and histological analyses (revised Supplementary Fig. 31). Per the reviewer's suggestion, we conducted additional investigations safety profile of Tri-NAb by evaluating its potential to induce cytokine release syndrome (CRS), characterized by excessive secretion of inflammatory cytokines. Following murine melanoma treatment, peripheral blood samples were collected to measure levels of specific inflammatory cytokines (including IL-10, TNF- α , IL-6, and IL-1 β). As depicted in Fig. R14 and revised Supplementary Fig. 37, the levels of these serum cytokines in Tri-NAb-treated mice did not significantly differ from those in untreated mice; thus, indicating that at the administered dose (2.5 mg/kg per mAb), Tri-NAb did not elicit CRS *in vivo*. The favorable biosafety and the absence of CRS may be attributed to the relatively low dosage of α 4-1BB administered in our study (for antitumor therapy, the dosage of α 4-1BB was 2.5 mg/kg, less than 60 μ g per mouse). Although the investigation of CRS occurrence following Tri-NAb administration was not performed on human PBMC, partly due to the high cost of human cytokine ELISA kits and NCG mice, we hope that the aforementioned data can alleviate reviewers' concerns regarding the safety of Tri-NAb. In case these findings fail to assuage his/her concerns, we are prepared to further supplement with relevant data utilizing human PBMC models and express our gratitude for the reviewer's understanding.

Fig. R14. The level of serum inflammatory cytokines. The levels of IL-10 (a), TNF- α (b),

IL-6 (c), and IL-1 β (d) in the peripheral blood of the treated mice were examined by ELISA. n.s., not significant. Data are presented as means \pm s.d. (n = 4). Statistical significance was calculated by one-way ANOVA with Tukey's post hoc test. *** $P < 0.001$.

Comment 3: What is ratio of the different antibody in the Tri-Nab being controlled to a consistency in each experiment? Is there a method to test the occupancy of each of the three antibodies? This has to be taken into consideration for the translational potential if the technology will be taken into use in the clinic.

Response: Thanks for the valuable comments. The equal probabilities of three mAbs binding to APCN@NA were confirmed at the outset of our study (data was not provided in the original manuscript). Tri-NAb was prepared by mixing APCN@NA with AF647-labeled α PDL1, AF488-labeled α 4-1BB, and AF555-labeled α NKG2A at a weight ratio of 1:1:1 at 4°C for 24 hours. The resulting Tri-NAb was effectively separated through high-speed centrifugation, while the exact quantity of unbound mAbs remaining in the supernatant was quantified HPLC, thereby enabling us to calculate the binding efficiency of each mAb to APCN@NA. As illustrated in Fig. R1 and Supplementary Fig. 16a, when the weight ratio between APCN@NA and total mAbs was 1:1, the binding efficiencies of α PDL1, α 4-1BB, and α NKG2A by APCN@NA were 89.5%, 91.6% and 91.8%, respectively. Based on these findings, it can be inferred that the final ratio of three mAbs on Tri-NAb was approximately 1:1:1. In the revised manuscript, we have revised relevant descriptions regarding the dosage of Tri-NAb.

Fig. R1. The binding efficiencies of α PDL1, α 4-1BB, and α NKG2A by APCN@NA, when the weight ratio between APCN@NA and total mAbs was 1:1. Data are presented as means \pm

s.d. (n = 3).

Comment 4: Is the APCN biodegradable? How stable is Tri-NAb? What is the molecular weight of the Tri-Nab? Are there any histological evidence that the Tri-Nab penetrate into tumor tissues?

Response: The APCN is composed of serum albumin and polylactic acid, both of which possess excellent biocompatibility and biodegradability properties. Serum albumin can be easily metabolized by the body, while polylactic acid is a commonly used biodegradable polymer material in the field of biomedicine (PMID: 31021482, PMID: 31681741). Based on this information, it is reasonable to presume that APCN could undergo degradation *in vivo*. Furthermore, we observed less than 6.0% release of α PDL1, α 4-1BB, and α NKG2A from Tri-NAb in PBS buffer over a period of 15 days, confirming its excellent stability (Fig. R15 and revised Supplementary Fig. 16b). Through asymmetric flow field-flow fractionation (AF4) and multi-angle light scattering (MALS) techniques provided by Wyatt Technology Corp., we measured the weight-average molecular weight of Tri-NAb to be 1.5×10^8 ($\pm 4.0\%$) g/mol with a radius of gyration (Rg) measuring at 54.1 ($\pm 4.1\%$) nm; these values were larger compared to those obtained for APCN with an Rg value of 37.4 ($\pm 2.4\%$) nm.

Fig. R15. Tri-NAb had good stability *in vitro*. Tri-NAb was prepared by mixing APCN@NA with AF647 labeled α PDL1, AF488 labeled α 4-1BB and AF555 labeled α NKG2A at 4°C for 24 h. Purified Tri-NAb was obtained by high-speed centrifugation and incubated in PBS buffer at 37°C. The released free antibodies were collected by high-speed centrifugation and quantified by fluorescence on days 0, 5, 10, and 15. Data are presented as means \pm s.d. (n = 3).

Further, a subcutaneous MC38 tumor model was established and Cy5-labelled NP_{IgG} or Cy5-labelled Tri-NAb was administered every three days for three repetitions after tumor inoculation. The injection dose of α PDL1, α 4-1BB, and α NKG2A was 2.5 mg/kg. Tumor tissues were harvested and frozen sections were made, which were then stained with DAPI (blue) and observed using CLSM. Through confocal microscopy (Fig. R16), it was observed that Tri-NAb equipped with three monoclonal antibodies penetrated the tumor tissue more effectively compared to NP_{IgG}. Related data was also included in Supplementary Fig. 28 in the revised manuscript.

Fig. R16. Representative fluorescent images of tumor lesions.

Comment 5: It is good that the authors presented PD-L1 expression in IFN γ -stimulated tumor cells *in vitro*, which is a given. Is there any data to demonstrate PD-L1 expression in tumors *in vivo* by IHC or IF to support the MOA of the Tri-Nab *in vivo*?

Response: Thanks for the meaningful comment. The positive expression of PD-L1 in B16-F10 melanoma tissues was examined using flow cytometry, as depicted in Fig. R17. Thus, with the assistance of Tri-NAb, tumor-infiltrating NK and CD8⁺ T cells can recognize and eliminate PDL1-positive tumor cells *in vivo*.

Fig. R17. Representative FACS histograms showing the expression of PDL1 in tumor tissue.

Comment 6: Supplement Figure 25 – NK1.1 separates NK very well as a clear population. Current gating does not represent typical NK cell population and not convincing that the population is NK cell. Authors need to present a more convincing gating data, such as CD3. Vs. NK1.1 to define NK population.

Response: Anti-tumor experiment of B16-F10 model was repeated and tumor tissues were collected at the end of treatment for flow cytometric analysis. The gating strategy recommended by the reviewer was employed to analyze tumor-infiltrating lymphocytes. As illustrated in Fig. R9, NK cell population was identified as CD45⁺CD3⁻NK1.1⁺, and Fluorescence Minus One (FMO) control for anti-NK1.1 was also included in the flow cytometric analysis. Additionally, CD45⁺CD3⁺ cells were gated to analyze CD8⁺ and CD4⁺ T cells. The revised gating strategy was included as Supplementary Fig. 35 in the revised Supplementary Information.

Fig. R9. Gating strategy for tumor-infiltrating lymphocyte.

Comment 7: Why *in vitro* data beautifully demonstrated the Tri-Nab brings tumor cells to simultaneously interact with NK and CD8 T cells, it would be more powerful to demonstrate the same interaction in tumors with multiplex IHC/IF.

Response: Thanks for the meaningful comment. Tri-NAb serves as a "bridge" *in vitro*, facilitating the interaction between NK/T cells and tumor cells by simultaneously

recognizing them. Per the reviewer's suggestion, a subcutaneous colon tumor model was established using green fluorescent protein (GFP)-expressing MC38 cells (MC38-GFP), and tumor-bearing mice were treated with IgG control, Tri-mAbs, NP_{αPDL1+α4-1BB}, NP_{αPDL1+αNKG2A}, NP_{α4-1BB+αNKG2A} or Tri-NAb (the doses of αPDL1, α4-1BB or αNKG2A: 2.5 mg/kg respectively) intravenously every three days to a total of three injections (q3dx3). The tumor tissues were harvested 24 hours post-last injection, sectioned, and stained with anti-NK1.1 and anti-CD8 antibodies for CLSM observation. As shown in Fig. R18, Tri-NAb treatment remarkably augmented the infiltration of both NK and CD8⁺ T cells compared to the other groups. Notably, both CD8⁺ T and NK cells were observed to be closely associated with tumor cells (indicated by white arrows), showcasing their heightened interaction with tumor cells. Related data was also included in Supplementary Fig. 29 in the revised manuscript.

Fig. R18. Representative confocal images of NK and CD8⁺ T cells interacting with MC38 tumor cells *in vivo*.

Comment 8: Supplement Fig 30b, what are the authors intending to demonstrate?

Response: Thanks for the kind reminder. Original Supplement Fig 30b demonstrated that Tri-NAb treatment did not elicit any hepatic or pulmonary injury, as confirmed by histological analyses. We have incorporated the pertinent description into the revised manuscript.

REVIEWERS' COMMENTS

Reviewer #1 (Remarks to the Author):

The authors have satisfactorily answered all the questions raised by the reviewers. The results now clearly support the conclusions made by the authors. These results demonstrate the efficacy of an interesting approach to modulate NK/CD8 T cells activation/inhibition by using nanoparticles.

Reviewer #2 (Remarks to the Author):

The authors have successfully addressed the concerns I had raised, thus as far as I am concerned the manuscript can be allowed to proceed to publication.

As a minor suggestion, grammatical proofreading of the manuscript is warranted prior to publication. For example on page 15, "Subsequently, though the T and B cell null Rag1-deficient (Rag1^{-/-}) mice, there was mild control over colon cancer growth following Tri-NAb treatment" would better be replaced by for example "A subsequent experiment in T and B cell null Rag1-deficient (Rag1^{-/-}) mice, revealed mild control over colon cancer growth following Tri-NAb treatment".

Reviewer #2 (Remarks on code availability):

There was not software code developed in this manuscript

Reviewer #3 (Remarks to the Author):

All concerns are adequately addressed with new data.

Reviewer #3 (Remarks on code availability):

The authors have adequately addressed all concerns.

Response to Reviewers' comments

Reviewer #1 (Remarks to the Author):

Comment: The authors have satisfactorily answered all the questions raised by the reviewers. The results now clearly support the conclusions made by the authors. These results demonstrate the efficacy of an interesting approach to modulate NK/CD8 T cells activation/inhibition by using nanoparticles.

Response: We express our profound gratitude for the reviewer's recognition of our diligent endeavors, and we extend our utmost appreciation to the reviewer for his/her insightful comments, which have significantly augmented the quality of our research.

Reviewer #2 (Remarks to the Author):

Comment: The authors have successfully addressed the concerns I had raised, thus as far as I am concerned the manuscript can be allowed to proceed to publication.

As a minor suggestion, grammatical proofreading of the manuscript is warranted prior to publication. For example, on page 15, "Subsequently, though the T and B cell null Rag1-deficient (Rag1^{-/-}) mice, there was mild control over colon cancer growth following Tri-NAb treatment" would better be replaced by for example "A subsequent experiment in T and B cell null Rag1-deficient (Rag1^{-/-}) mice, revealed mild control over colon cancer growth following Tri-NAb treatment".

Response: We express our profound gratitude for the reviewer's recognition of our diligent endeavors, and we extend our utmost appreciation to the reviewer for his/her insightful comments, which have significantly augmented the quality of our research.

Following the reviewer's suggestion and in order to avoid any potential grammatical errors, we have sought assistance from subject-expert editors from Springer Nature Author Services for proofreading the language in our manuscript.

Reviewer #3 (Remarks to the Author):

Comment: All concerns are adequately addressed with new data.

Response: We express our profound gratitude for the reviewer's recognition of our diligent endeavors, and we extend our utmost appreciation to the reviewer for his/her insightful comments, which have significantly augmented the quality of our research.